# Comparative therapeutic efficacy of remdesivir and combination lopinavir, ritonavir, and interferon beta against MERS-CoV

Timothy P. Sheahan [1,5]*, Amy C. Sims[1,5], Sarah R. Leist[1], Alexandra Schäfer[1], John Won[1], Ariane J. Brown[1], Stephanie A. Montgomery [2], Alison Hogg[3], Darius Babusis [3], Michael O. Clarke[3], Jamie E. Spahn[3], Laura Bauer[3], Scott Sellers[3], Danielle Porter[3], Joy Y. Feng[3], Tomas Cihlar[3], Robert Jordan[3], Mark R. Denison [4] & Ralph S. Baric[1]*

Middle East respiratory syndrome coronavirus (MERS-CoV) is the causative agent of a severe respiratory disease associated with more than 2468 human infections and over 851 deaths in 27 countries since 2012. There are no approved treatments for MERS-CoV infection although a combination of lopinavir, ritonavir and interferon beta (LPV/RTV-IFNb) is currently being evaluated in humans in the Kingdom of Saudi Arabia. Here, we show that remdesivir (RDV) and IFNb have superior antiviral activity to LPV and RTV in vitro. In mice, both prophylactic and therapeutic RDV improve pulmonary function and reduce lung viral loads and severe lung pathology. In contrast, prophylactic LPV/RTV-IFNb slightly reduces viral loads without impacting other disease parameters. Therapeutic LPV/RTV-IFNb improves pulmonary function but does not reduce virus replication or severe lung pathology. Thus, we provide in vivo evidence of the potential for RDV to treat MERS-CoV infections.

[1] Department of Epidemiology, University of North Carolina at Chapel Hill, Chapel Hill, NC, USA. [2] Department of Pathology & Laboratory Medicine, University of North Carolina, Chapel Hill, NC, USA. [3] Gilead Sciences, Inc, Foster City, CA, USA. [4] Department of Pediatrics-Infectious Diseases, Department of Pathology, Microbiology and Immunology, Vanderbilt University Medical Center, Nashville, TN, USA. [5] These authors contributed equally: Timothy P. Sheahan, Amy C. Sims. *email: sheahan@email.unc.edu; rbaric@email.unc.edu

The coronavirus (CoV) family has a propensity for emergence into new hosts often causing novel severe disease. In 2012, Middle East respiratory syndrome coronavirus (MERS-CoV), was discovered as the causative agent of a severe respiratory syndrome in the Kingdom of Saudi Arabia (KSA), has since caused at least 2468 cases and 851 deaths globally[1]. MERS-CoV is endemic in camels, the zoonotic reservoir host, with evidence of infection going back at least 30 years[2]. Camels in the Middle East and perhaps in East Africa continue to seed human infections which may require hospitalization especially in aged individuals with preexisting comorbidities[1,3,4]. Similar to severe acute respiratory syndrome CoV (SARS-CoV), MERS-CoV has spread to over 27 countries via air travel of infected people[5]. In 2014, a single imported case caused an outbreak of 186 cases in South Korea, while a more recent case imported from the Middle East was contained as a result of rapid implementation of public health measures[6]. MERS-CoV continues to cause human infections globally and thus is listed as a priority pathogen with pandemic potential by World Health Organization (WHO) and the Coalition for Epidemic Preparedness Innovations (CEPI). Presently, there are no approved treatments for MERS-CoV or any other human CoV.

Emerging viral diseases typically have very few if any effective treatment options. As such, treatments designed and approved for other diseases are administered to patients with emerging viral syndromes empirically based on limited clinical or laboratory data. Multiple U.S. Food and Drug Administration (FDA) approved therapies have been evaluated for antiviral activity against MERS-CoV in vitro including lopinavir (LPV), ritonavir (RTV), and interferon beta (IFNb). LPV is a human immunodeficiency virus 1 (HIV-1) protease inhibitor that is usually combined with RTV to increase LPV half-life through the inhibition of cytochrome P450[7]. Although the antiviral activity of LPV against MERS-CoV has been reported in Vero cells (concentration causing a 50% reduction in replication ($EC_{50} = 8\,\mu M$)), other studies report complete inactivity similar to that of RTV[8,9]. In contrast, studies evaluating the antiviral activity of type I and type II interferons have reported IFNb as the most potent interferon ($EC_{50}$ 1.37–17 IU/mL) in reducing MERS-CoV replication in vitro[8,10]. The only in vivo study assessing the therapeutic efficacy of LPV/RTV or IFNb against MERS-CoV published thus far was performed in common marmosets where modest improvements in clinical outcomes were noted[11]. In human MERS-CoV patients, two published case reports describe conflicting results on the use of a combination of LPV/RTV, pegylated interferon, and ribavirin with one of two patients surviving[12,13]. To this end, a randomized control trial (MIRACLE Trial) aimed at conclusively determining if LPV/RTV-IFNb improves clinical outcomes in MERS-CoV patients was initiated in 2016 and has thus far enrolled 76 patients in KSA[14,15].

Remdesivir (RDV, GS-5734) is a broad-spectrum antiviral nucleotide prodrug with potent in vitro antiviral activity against a diverse panel of RNA viruses such as Ebola virus (EBOV), Marburg, MERS-CoV, SARS-CoV, respiratory syncytial virus (RSV), Nipah virus (NiV), and Hendra virus[16–18]. The mechanism of RDV's anti-MERS-CoV activity is likely through premature termination of viral RNA transcription as shown in biochemical assays using recombinant EBOV, NiV, and RSV polymerases[18–20]. In primary human lung epithelial cell cultures, RDV is potently antiviral against circulating contemporary human CoVs, SARS-CoV ($EC_{50} = 0.07\,\mu M$), MERS-CoV ($EC_{50} = 0.07\,\mu M$), and related zoonotic bat CoVs[17,21]. We recently reported that therapeutic RDV improves disease outcomes and reduces viral loads in SARS-CoV-infected mice[17]. Since similar studies had not been performed with MERS-CoV, we generated a transgenic mouse with a humanized MERS-CoV receptor

(dipeptidyl peptidase 4, hDPP4) and deleted for carboxylesterase 1c (Ces1c) to improve the pharmacokinetics of nucleotide prodrugs such that it better approximates the drug exposure profile in humans[22]. Here, we show that RDV provides superior antiviral activity against MERS-CoV in vitro and in vivo as compared with LPV/RTV-IFNb. In addition, RDV was the only therapeutic treatment to significantly reduce pulmonary pathology. Thus, we provide in vivo evidence of the potential for RDV to treat MERS-CoV infections.

## Results

**RDV and IFNb have superior antiviral activity to LPV and RTV.** We utilized a recombinant MERS-CoV engineered to express a reporter nanoluciferase (MERS-nLUC) for our in vitro antiviral activity assays. To ensure that our reporter virus behaved similarly to WT MERS-CoV, we first demonstrated that MERS-nLUC and wild-type (WT) MERS-CoV EMC 2012 strain replicate to similar levels in the absence of drug treatment and are similarly susceptible to the antiviral activity of RDV (WT $EC_{50} = 0.12\,\mu M$; MERS-nLUC $EC_{50} = 0.09\,\mu M$) in the human lung epithelial cell line, Calu-3 (Supplementary Fig. 1). These data are in agreement with those for WT MERS Jordan strain ($EC_{50} = 0.3\,\mu M$) reported by Warren et al.[18]. Thus, future reporter virus data should be representative of WT MERS-CoV.

We then performed parallel antiviral assays in Calu-3 cells with MERS-nLUC comparing LPV, RTV, IFNb, and RDV (Fig. 1; Supplementary Fig. 2)[17]. Similar to our previous reports, RDV showed potent inhibition of MERS-CoV replication with a $EC_{50}$ of $0.09\,\mu M$, no observable cytotoxicity up to $10\,\mu M$ and a selectivity index ($SI = EC_{50}/CC_{50}$) > 100[17]. (Fig. 1a). In contrast, the respective $EC_{50}$ values generated for LPV and RTV were 11.6 and $24.9\,\mu M$ with $CC_{50}$ values >50 µM (Fig. 1b, c). Thus, the SI for LPV and RTV was > 4.3 and > 2, respectively. Combination LPV and RTV (LPV:RTV, 4.6:1 molar ratio) is currently under evaluation in the MIRACLE trial[14]. The antiviral activity of LPV/RTV ($EC_{50} = 8.5\,\mu M$) was similar to LPV alone ($EC_{50} = 11.6\,\mu M$, $P = 0.43$, Wilcoxon matched-pairs signed rank test), suggesting the effect was largely driven by LPV (Fig. 1d). We found potent inhibition of MERS-CoV with IFNb ($EC_{50} = 175$ international units (IU)/mL) (Fig. 1e), $CC_{50}$ values >2800 IU/mL and an SI > 16. Together, these data demonstrate that RDV and IFNb have superior in vitro antiviral activity compared with LPV and RTV, and that RTV does not significantly enhance the antiviral activity of LPV in vitro.

**IFNb activity is not improved when combined with LPV/RTV.** Cell culture medium and human plasma have different concentrations and types of proteins, which directly affects the levels of biologically available free drug unbound to protein complicating the comparison of drug levels the respective systems[23]. To address this issue, we utilized comparative equilibrium dialysis (CED) to determine the differences in free unbound drug between human plasma and cell culture medium revealing that the maximal plasma concentration ($C_{max}$) in human plasma (15 µM) had the same amount of free unbound LPV as 5 µM LPV in 10% FBS containing cell culture medium. Thus, we combined LPV (5 µM) and RTV (1.09 µM) at a fixed molar ratio of 4.6:1 in combination with increasing concentrations of IFNb. The antiviral activity of the LPV/RTV-IFNb combination ($EC_{50} = 160$ IU/mL) was indistinguishable from that of IFNb alone ($EC_{50} = 175$ IU/mL) ($P = 0.62$, Wilcoxon matched-pairs signed rank test) (Fig. 1f). These data suggest that the observed in vitro antiviral activity of the LPV/RTV-IFNb combination on MERS-CoV is dominated by IFNb when LPV/RTV is used at clinically relevant concentrations.

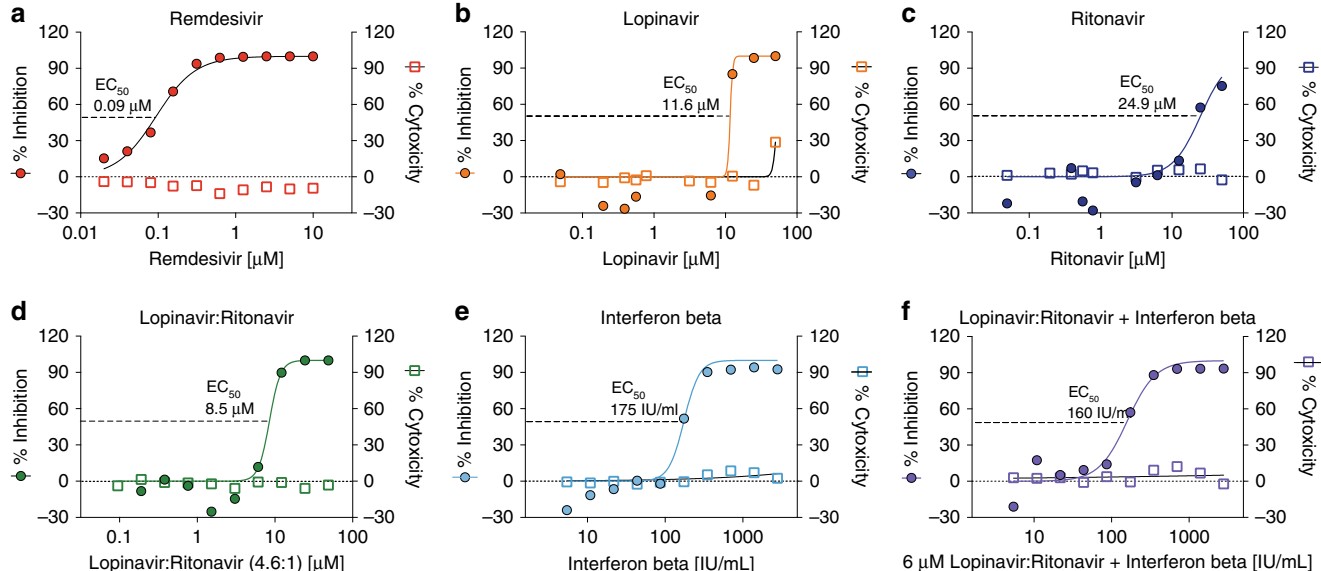

**Fig. 1 RDV and IFNb have superior antiviral activity to LPV and RTV.** Graphs depict mean % inhibition of MERS-CoV replication (left $Y$-axis) and % cytotoxicity (right $Y$-axis) of antivirals. Calu-3 cells were infected in sextuplicate with MERS-CoV nanoluciferase (nLUC) at a multiplicity of infection (MOI) of 0.08 in the presence of a dose response of drug for 48 h, after which replication was measured through quantitation of MERS-CoV–expressed nLUC. Cytotoxicity was measured in similarly treated but uninfected cultures via Cell-Titer-Glo assay. Representative data are shown from four independent experiments.

**A MERS-CoV mouse model for the testing of nucleotide prodrugs.** Unlike humans, mice have high levels of a serum esterase (carboxylesterase 1c, *Ces1c*) that drastically reduces the stability of RDV in mice requiring efficacy studies be performed in *Ces1c*$^{-/-}$ mice to better approximate the pharmacokinetics and drug exposure profile in humans[17]. MERS-CoV infection of standard laboratory mice is prevented due to differences in human and mouse dipeptidyl peptidase 4 (DPP4), the entry receptor for MERS-CoV. To enable testing of RDV in mice, we bred *Ces1c*$^{-/-}$ mice with mice harboring a modified *DPP4* humanized via CRISPR/Cas9 at residues 288 and 330 (*hDPP4*). The resultant *Ces1c*$^{-/-}$ *hDPP4* mice had indistinguishable virus replication and pathogenesis from *hDPP4* mice when infected with MERS-CoV (Supplementary Fig. 3)[22,24].

**Prophylactic RDV diminishes MERS-CoV replication and disease.** Using the new *Ces1c*$^{-/-}$ *hDPP4* mouse model, we sought to determine if prophylactic RDV could ameliorate MERS-CoV disease. As shown in Fig. 2a, prophylactic RDV (25 mg/kg, BID) administered 1 day prior to infection significantly diminished MERS-CoV-induced weight loss in mice infected with 5E + 04 ($P < 0.0001$, two-way ANOVA with Tukey's multiple comparison test) or 5E + 05 plaque-forming units (pfu) ($P < 0.0001$, two-way ANOVA with Tukey's multiple comparison test) as compared with similarly infected vehicle-treated animals. Moreover, RDV administered prior to infection also prevented mortality ($P = 0.0037$, Mantel–Cox test) in those administered a lethal dose (i.e., 5E + 05 pfu) (Fig. 2b). In contrast to vehicle-treated animals, lung hemorrhage was significantly reduced ($P < 0.0001$, two-way ANOVA with Tukey's multiple comparison test) with RDV prophylaxis (Fig. 2c)[22,25,26]. Importantly, RDV prophylaxis significantly reduced virus lung titers > 3 logs on both 4 (5E + 04 $P = 0.0240$, 5E + 04 $P = 0.0001$, two-way ANOVA with Sidek's multiple comparison test) and 6 days post infection (dpi) (5E + 05 $P = 0.0001$, two-way ANOVA with Sidek's multiple comparison test) (Fig. 2d), which was corroborated by viral antigen labeling in lung tissue sections (Fig. 2e).

**RDV prophylaxis reduces features of ALI.** The American Thoracic Society (ATS) issued a consensus document with defined parameters and tools to more accurately translate small animal models of acute lung injury (ALI) to the human condition[26]. Using the ATS Lung Injury Scoring System designed to quantitate histopathological features of ALI (Fig. 3a), we blindly scored fields of hematoxylin and eosin stained lung tissue sections from the mice in Fig. 2 for the following features: neutrophils in the alveolar and interstitial space, hyaline membranes, proteinaceous debris filling the air spaces, and alveolar septal thickening[26]. As compared with control mice, the ATS lung injury scores were significantly reduced in RDV-treated mice ($P = 0.0079$, Mann–Whitney test) infected with 5E + 04 pfu MERS-CoV and approached statistical significance ($P = 0.0536$, Mann–Whitney test) in those infected with the higher virus dose (5E + 05 pfu) (Fig. 3a). In Fig. 3, we also show examples of the quantitated pathological features. In normal healthy mock-infected control mice (Fig. 3b), the alveolar air spaces are free of debris and inflammatory cells, the walls (i.e., septae) of the alveolar sac are thin which facilitates efficient gas exchange and rare neutrophils in circulation are seen in the capillaries within alveolar septae, but not in the air spaces. In contrast, vehicle-treated MERS-CoV-infected animals in both viral dosage groups (Fig. 3c) had multiple histologic features of ALI including notable immune cell infiltration into the alveolar septae and resultant septal wall thickening, scattered degenerating and dying cells, proteinaceous debris in the air spaces resulting from capillary leakage some of which is organized into hyaline membranes and neutrophils in alveolar septae as well as in the air spaces. Interestingly, the early hyaline membranes (i.e., thin pink material lining alveoli) were noted in vehicle-treated animals infected with the lower dose of MERS-CoV (Fig. 3c, left), while more developed hyaline membranes (i.e., thick pink material lining alveoli) were observed in vehicle-treated animals infected with the higher dose of virus (Fig. 3c, right). Importantly, the air spaces of RDV-treated animals infected with the lower dose of MERS-CoV (Fig. 3d, left) predominantly lacked cellular debris, immune cells, and hyaline membranes although alveolar septal thickening due to increased

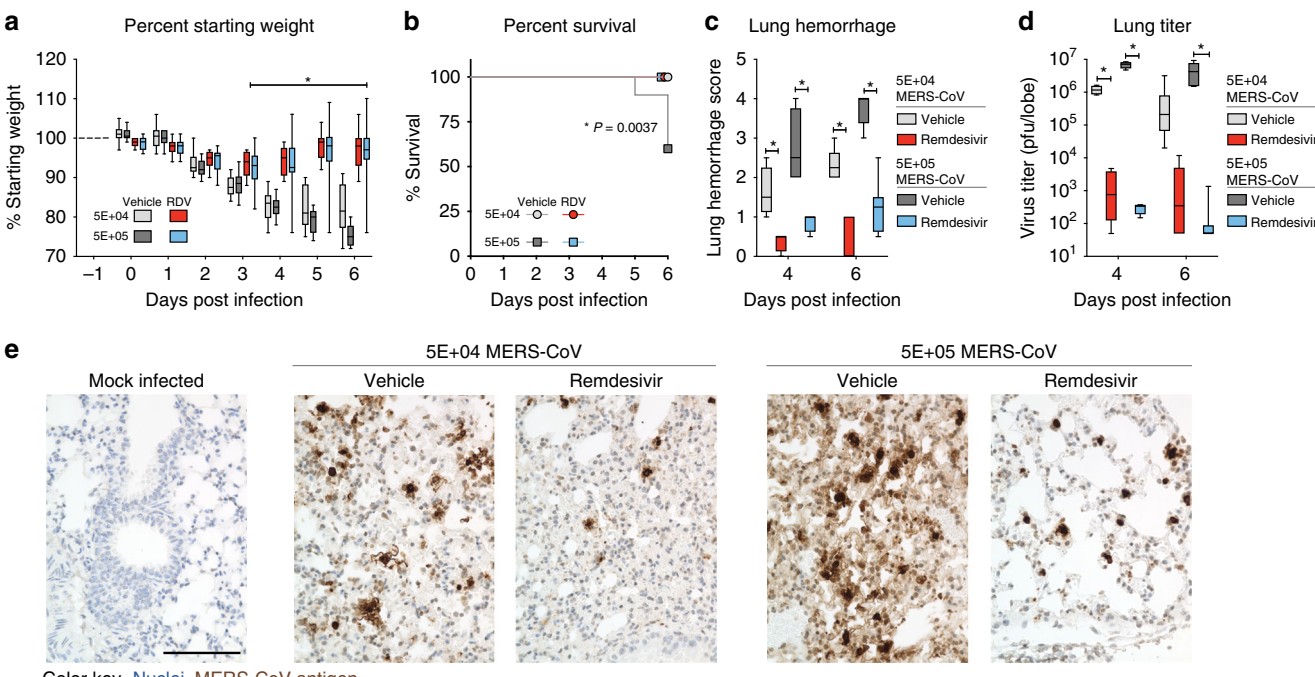

**Fig. 2 Prophylactic RDV reduces MERS-CoV replication and disease. a** Percent starting weight of 9–12-week-old male and female $Ces1c^{-/-}$ $hDPP4$ mice prophylactically administered subcutaneous vehicle or remdesivir (RDV, 25 mg/kg) BID the day prior to infection with either 5E + 04 (vehicle $n = 14$, RDV $n = 14$) or 5E + 05 (vehicle $n = 14$, RDV $n = 15$) plaque-forming units (pfu) MERS M35C4. Asterisks indicate statistically significant differences ($P < 0.05$) as determined by two-way ANOVA and Tukey's multiple comparison test. **b** Percent survival of each cohort and survival analysis by Mantel–Cox test ($P < 0.05$, N per group noted in **a**). **c** Lung hemorrhage scored on a scale of 0–4, where 0 is a normal pink healthy lung and 4 is a completely dark red lung. On 4 dpi, $N = 4$/group, and on 6 dpi the remaining animals are plotted. Asterisks indicate statistically significant differences ($P < 0.05$) as determined by two-way ANOVA and Tukey's multiple comparison test. **d** MERS-CoV lung titer on 4 ($N = 4$) and 6 dpi (all remaining animals). Asterisks indicate statistically significant differences ($P < 0.05$) as determined by two-way ANOVA and Sidek's multiple comparison test. For **a**, **c**, **d**, the boxes encompass the 25th to 75th percentile, the line is at the median, while the whiskers represent the range. **e** Hematoxylin (nuclei, blue) and immunostaining for MERS-CoV antigen (brown) in lung tissue sections from 4 dpi. All photos were taken with the same magnification. The black bar indicates 100 μM scale. Images from representative mice for each group are shown.

immune cells was observed. RDV-treated animals infected with the higher dose of MERS-CoV (Fig. 3d, right) similarly lacked debris and inflammatory infiltrates in the air spaces, but exhibited increased alveolar septal thickening. Thus, our blind pathological evaluation of lung tissue sections using the ATS Lung Injury Scoring System demonstrates that prophylactic RDV diminished the pathological features of ALI in MERS-CoV-infected mice[26].

**The minimal effect of prophylactic LPV/RTV-IFNb on MERS-CoV.** Prophylactic studies provide a best case scenario to evaluate in vivo antiviral activity since time is given for the metabolization and accumulation of antiviral agents within cells targeted by virus prior to infection. To determine if prophylactic LPV/RTV-IFNb improved outcomes following MERS-CoV infection, we first confirmed that subcutaneous human equivalent doses (H.E.D.) of IFNb exerted a biological effect in mice. Administration of a 1×, 2.5x, or 25x H.E.D. of mouse IFNb rapidly induced dose-dependent expression of interferon stimulated gene (ISG), $Mx1$, in peripheral blood mononuclear cells (PBMCs) in $Ces1c^{-/-}$ mice (Supplementary Fig. 4a). Importantly, the kinetics of $Mx1$ gene expression in target organ of MERS-CoV, the lung, was similar in both $Ces1c^{-/-}$ and $Ces1c^{-/-}$ $hDPP4$ mice (Supplementary Fig. 4b). Similarly, a single dose of IFNb significantly induced sustained expression ($P < 0.05$, two-way ANOVA with Sidek's multiple comparison test) of interferon gamma-induced protein 10 (IP-10, CXCL-10) in the serum of both strains of mice (Supplementary Fig. 4c). Thus, we observed an expected

ISG response in the blood and lung tissue of both $Ces1^{-/-}$ and $Ces1^{-/-}$ $hDPP4$ mice following IFNb treatment.

We then evaluated if LPV/RTV-IFNb prophylaxis could improve outcomes in $Ces1^{-/-}$ $hDPP4$ mice infected with 5E + 04 pfu MERS-CoV. We compared vehicle (oral: propylene glycol and ethanol, subcutaneous: PBS) to three different treatment scenarios, including LPV/RTV-IFNb high (25x H.E.D. IFNb), LPV/RTV-IFNb low (1x H.E.D. IFNb), or IFNb-high alone (25x H. E.D. IFNb) (Fig. 4). Prophylactic RDV and vehicle were included as controls. Since ISG expression peaks 2–4 h after IFNb administration (Supplementary Fig. 4), we initiated IFNb dosing 2 h prior to MERS-CoV infection to maximize the potential antiviral effect. Similar to our previous studies, RDV (25 mg/kg, BID) or vehicle were administered subcutaneously every 12 h to obtain exposures in mice similar to that observed in humans[17]. The dosing levels and frequencies of LPV/RTV (oral once daily) and IFNb (subcutaneously every other day) were chosen to mirror those in the MIRACLE trial[27]. Unlike RDV-treated mice (Fig. 4a), vehicle, LPV/RTV-IFNb, or IFNb alone did not prevent weight loss (Fig. 4b). In fact, animals administered IFNb alone lost significantly more weight than vehicle ($P = 0.01$), LPV/RTV-IFNb high ($P = 0.0001$) and low ($P = 0.006$, all two-way ANOVA with Tukey's multiple comparison test) groups (Fig. 4b). On 6 dpi, RDV treatment reduced virus lung titers the most (>3 log reduction, vehicle median = 1.4E + 05 pfu/lobe, RDV median = 50 pfu/lobe) (Fig. 4a), while those in LPV/RTV-IFNb low treated were modestly reduced (~4-fold, median = 5.5E + 03 pfu/lobe) as compared with its vehicle (median = 2.3E + 04 pfu/lobe) ($P = 0.04$, two-way

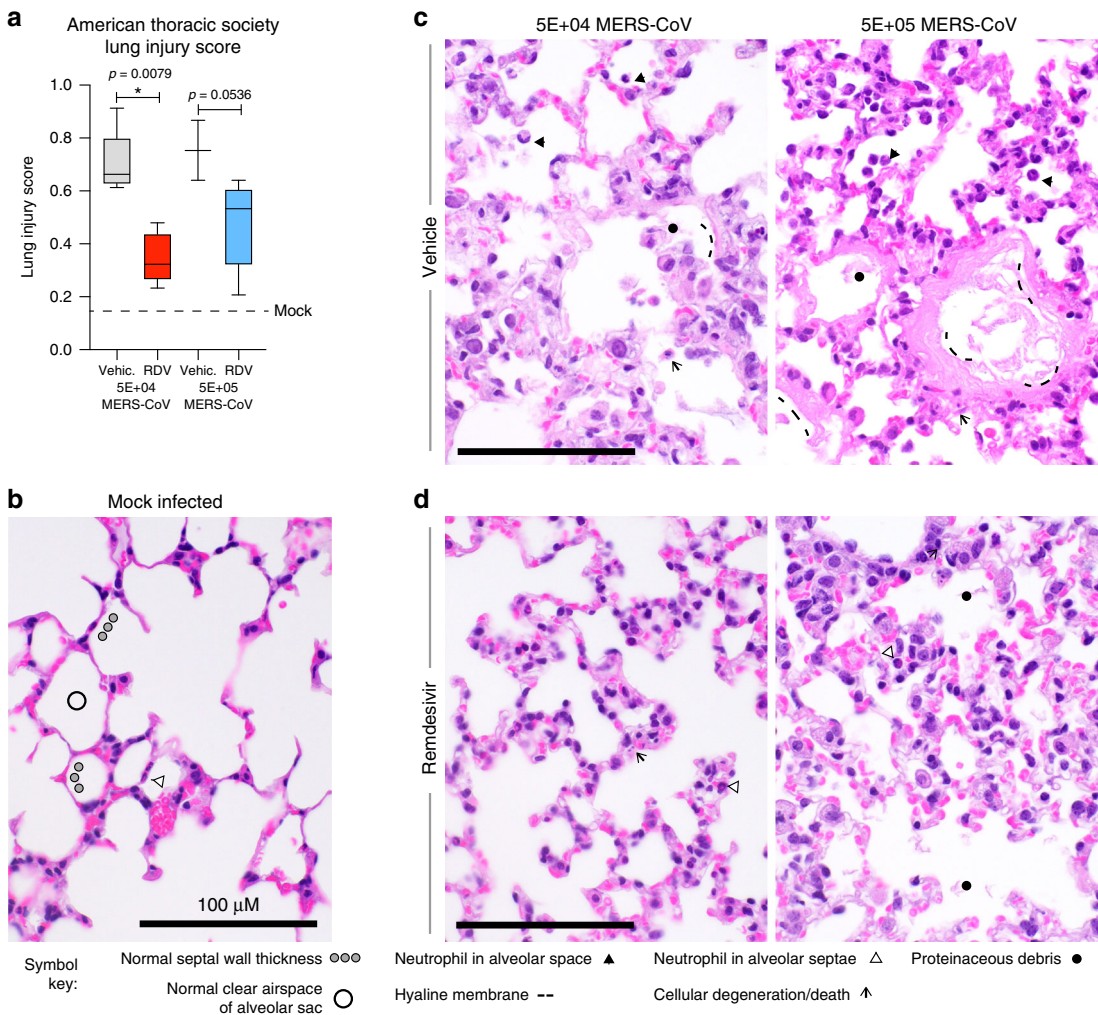

**Fig. 3 Remdesivir prophylaxis reduces features of acute lung injury.** The histological features of acute lung injury were blindly scored using the American Thoracic Society Lung Injury Scoring system creating an aggregate score for the following phenotypes: neutrophils in the alveolar and interstitial space, hyaline membranes, proteinaceous debris filling the air spaces, and alveolar septal thickening. Three randomly chosen high power (×60) fields of diseased lung were assessed per mouse. Representative images are shown for mock infected as well as those administered prophylactic vehicle or RDV and infected with either 5E + 04 or 5E + 05 pfu mouse adapted MERS-CoV. The numbers of mice scored per group: Vehicle 5E + 04 pfu MERS-CoV N = 5, Vehicle 5E + 05 pfu MERS-CoV N = 3, RDV 5E + 04 pfu MERS-CoV N = 5, RDV 5E + 05 pfu MERS-CoV N = 5. Symbols identifying example features of disease are indicated in the figure. All images were taken at the same magnification. The black bar indicates 100 μm scale. For the graph, the boxes encompass the 25th to 75th percentile, the line is at the median, while the whiskers represent the range. Statistical significance was determined by Mann–Whitney test.

ANOVA with Sidek's multiple comparison test) (Fig. 4b). Upon measuring pulmonary function by whole-body plethysmography (WBP), we found only RDV prophylaxis improved pulmonary function (Fig. 4a, b) (P = 0.002 to 0.0001, two-way ANOVA with Sidek's multiple comparison test). Interestingly, animals receiving IFNb-alone had significantly worse pulmonary function than companion groups at later times post infection (4–5 dpi, P = < 0.05, two-way ANOVA with Tukey's multiple comparison test). To determine if initiation time or dose level would affect outcomes in the IFNb only group, we performed a similar study but administered IFNb at a lower dose (i.e., 1x H.E.D.) 24 h prior to infection (Supplementary Fig. 5). Alteration of IFNb dose or time of administration did not improve outcomes from those in Fig. 4, and the modest improvement in virus titer seen with LPV/RTV-IFNb noted above was not observed. Taken together, prophylactic LPV/RTV-IFNb caused modest reductions in lung viral load in one of two studies, but had minimal impact on other disease metrics while IFNb alone did not impact virus replication and exacerbated disease.

**Therapeutic RDV reduces MERS-CoV replication and pathology.** To more stringently assess the therapeutic potential of RDV and LPV/RTV-IFNb and to better model the human scenario where MERS-CoV patients most likely would initiate treatment after infection, we performed a series of therapeutic efficacy studies in mice. We initiated the following treatments in $Ces1^{-/-}$ hDPP4 mice infected with 5E + 04 pfu MERS-CoV on 1 dpi: RDV or vehicle, LPV/RTV-IFNb low (1× human equivalent), LPV/RTV-IFNb high (25× human equivalent) or their vehicles. Dose route, amount, and frequency were similar to the prophylactic studies above. Only therapeutic RDV substantially reduced body weight loss (P = 0.019 to < 0.0001, two-way ANOVA with Tukey's multiple comparison test) (Fig. 5a, b) and lung hemorrhage on 6 dpi (Fig. 5c) (P < 0.0001, one-way ANOVA with Kruskal–Wallis test). Similarly, only RDV treatment significantly reduced virus lung titers on 6 dpi (vehicle median 7.8E + 05 pfu/lobe, RDV median 125 pfu/lobe, P = 0.0001, one-way ANOVA with Kruskal–Wallis test) (Fig. 5d), which we corroborated with viral antigen labeling in lung tissue sections (Fig. 5e). Similar

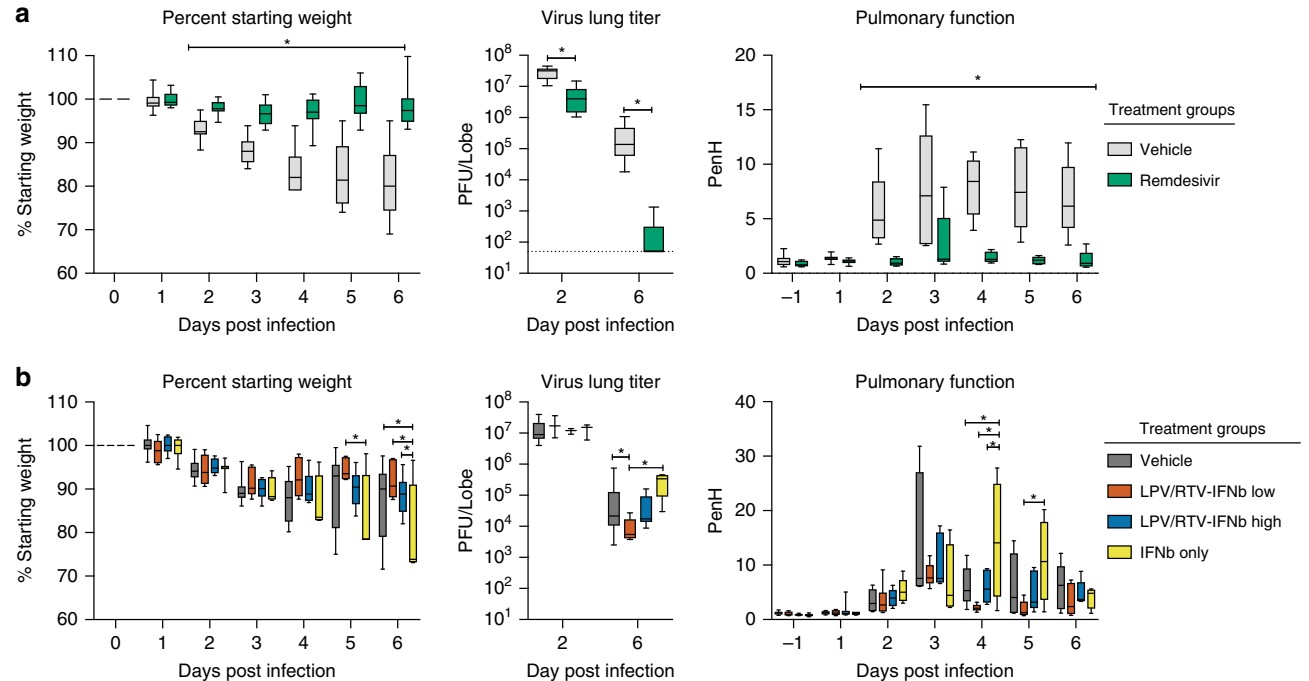

**Fig. 4 Prophylactic LPV/RTV + IFNb does not improve disease outcomes. a** Percent starting weight (Left) of 12–14-week-old female *Ces1c*−/− *hDPP4* mice infected with 5E + 04 pfu MERS M35C4 and treated BID with either vehicle (*n* = 9) or remdesivir (RDV, 25 mg/kg, *n* = 9) subcutaneously beginning −1 dpi. Asterisks indicate statistically significant differences (*P* < 0.05) as determined by two-way ANOVA and Tukey's multiple comparison test. (Middle) MERS-CoV lung titer on 2 (*N* = 3) and 6 dpi (all remaining animals). Asterisks indicate statistically significant differences (*P* < 0.05) as determined by Mann–Whitney test. (Right) WBP was used to assess pulmonary function in mice. PenH is a surrogate measure of airway resistance or bronchoconstriction. Asterisks indicate statistical differences by two-way ANOVA with Sidek's multiple comparison test. **b** Percent starting weight (left), virus lung titer (middle), and pulmonary function metric PenH (right) of cohorts of mice similar in age and sex and infected similarly with MERS-CoV as in **b** but treated with vehicle (*n* = 9), LPV/RTV + IFNb low (1× human equivalent) (*n* = 9), LPV/RTV + IFNb high (25× human equivalent) (*n* = 9), or IFNb high only (*n* = 9). Oral vehicle or lopinavir/ritonavir (160/40 mg/kg) were administered orally once daily beginning the −1 dpi. IFNb treatment was initiated 2 h prior to infection and every other day thereafter. To control for dosing effects, vehicle-treated mice received both LPV/RTV vehicle and subcutaneous PBS to mirror IFNb injections. Likewise, IFNb only group received oral vehicle to mirror that seen in orally dosed groups. Similar statistical tests performed on **a** were performed on **b**. For the box and whisker plots, the boxes encompass the 25th to 75th percentile, the line is at the median, while the whiskers represent the range.

therapeutic studies (Supplementary Fig. 6) were performed but with a lethal dose of MERS-CoV (5E + 05 pfu) where no treatment improved survival (Supplementary Fig. 6a, d) or lung hemorrhage (Supplementary Fig. 6b, e), but therapeutic RDV significantly reduced lung viral load on 6 dpi (Supplementary Fig. S6c) (*P* = 0.03, Mann–Whitney test). Thus, therapeutic LPV/RTV-IFNb failed to improve weight loss, lung hemorrhage, and virus lung titer after 5E + 04 pfu MERS-CoV and did not improve survival following a lethal dose of MERS-CoV (Supplementary Fig. 6d–f). In contrast, therapeutic RDV diminished weight loss, lung hemorrhage, and virus replication during an ongoing MERS-CoV infection, but the degree of clinical benefit is dependent on viral dose and time of treatment initiation.

**Therapeutic RDV and LPV/RTV-IFNb improve pulmonary function.** In the therapeutic antiviral efficacy studies described in Fig. 5, we used WBP to assess pulmonary function (Fig. 6). In contrast to vehicle-treated animals, RDV-treated animals had reduced flow rate at 50% of the expired volume (EF50) (*P* = 0.01, two-way ANOVA with Sidek's multiple comparison test), and PenH (*P* = 0.04, two-way ANOVA with Sidek's multiple comparison test), a surrogate measure of airway resistance/obstruction (Fig. 6a)[28]. Similarly, Rpef (the fraction of expiration before peak expiratory flow is reached), an indicator of bronchoconstriction, returned to baseline in RDV-treated animals by 5 dpi (*P* = 0.002, two-way ANOVA with Sidek's multiple comparison

test) yet remained suppressed in vehicle-treated animals. Unlike the prophylactic study in Fig. 4b, therapeutic LPV + RTV-IFNb low improved pulmonary function as compared with vehicle with significantly reduced EF50 and PenH (*P* < 0.05, two-way ANOVA with Tukey's multiple comparison test) and baseline levels of Rpef by 5 dpi (*P* < 0.05, two-way ANOVA with Tukey's multiple comparison test) (Fig. 6b). While therapeutic LPV/RTV-IFNb failed to reduce weight loss, lung hemorrhage, and virus titer, this regimen appears to provide improvements in pulmonary function similar to therapeutic RDV.

**Therapeutic RDV but not LPV/RTV-IFNb diminishes signs of ALI.** We then quantitated the lung pathology in therapeutically treated animals (Fig. 7). In lung tissue sections from the 6 dpi, we blindly evaluated and scored features of ALI using three different and complementary approaches. In Fig. 7a, we show reduced ALI with RDV treatment. Although sporadic apoptotic cells and alveolar septal wall thickening (~2–4-fold over mock) driven by immune cell infiltration were observed in the lungs of mice treated with therapeutic RDV, the air spaces remained predominantly free of cellular debris and inflammatory cells similar to mock-infected animals (Fig. 7a). In contrast, multiple pathological features typical of ALI were noted in both vehicle groups (RDV and LPV/RTV vehicles) as well as both LPV/RTV-IFNb low and high, including altered alveolar architecture due to pneumocyte degeneration and death, numerous inflammatory

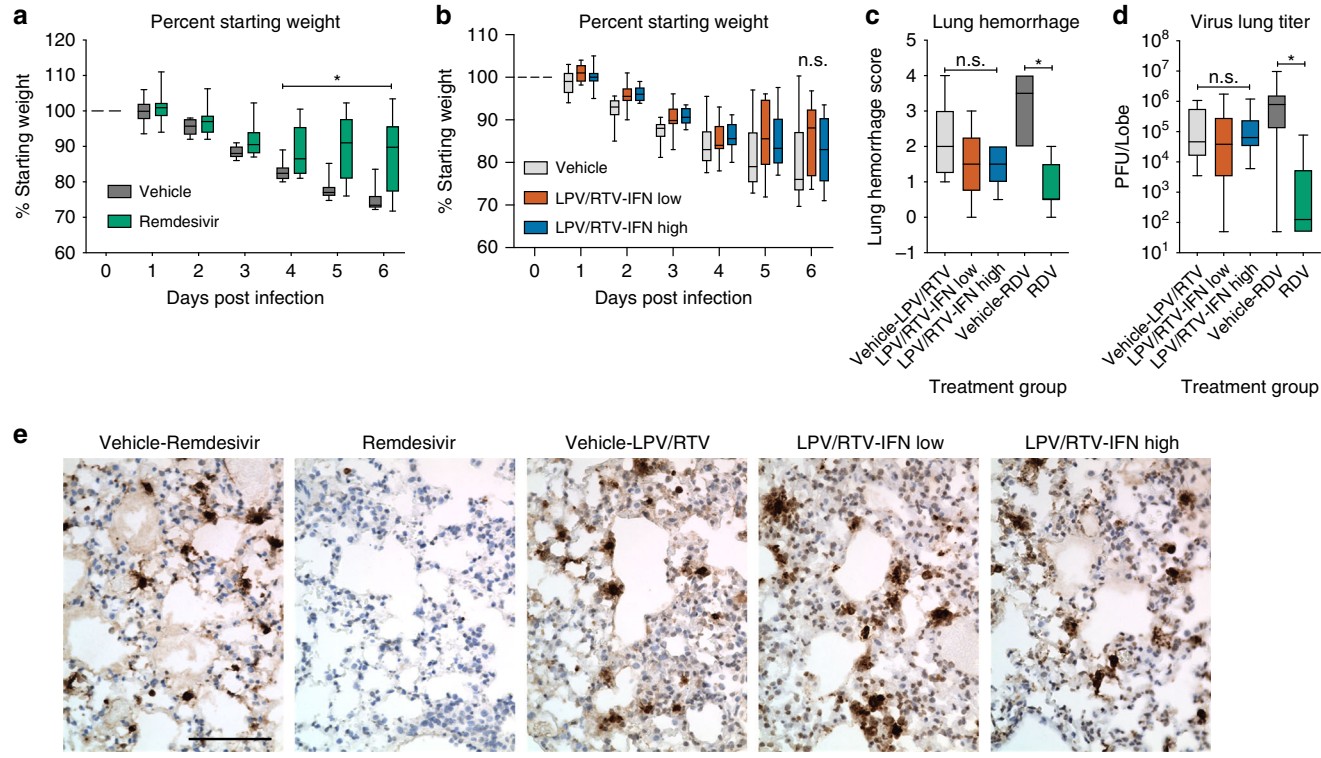

**Fig. 5 Therapeutic RDV reduces replication and pathology.** Percent starting weight of 10–12-week-old female *Ces1c*−/− *hDPP4* mice infected with 5E + 04 pfu MERS M35C4 and treated with **a** subcutaneous vehicle for RDV (*N* = 13) or remdesivir (RDV, 25 mg/kg, *N* = 14) BID beginning 1 dpi or **b** vehicle for LPV/RTV-IFNb (*N* = 15), LPV/RTV-IFNb low (*N* = 16) or LPV/RTV-IFNb high (*N* = 16) beginning 1 dpi. Oral vehicle or lopinavir/ritonavir (160/40 mg/kg) was administered orally once daily. IFNb low (1x human equivalent dose of 1.6 MIU/kg) and high (25x human equivalent dose of 40 MIU/kg) or PBS vehicle were administered via subcutaneous injection every other day. Asterisks indicate statistical differences by two-way ANOVA with Tukey's multiple comparison test. **c** Lung hemorrhage 6 dpi for all animals in **a**, **b** scored on a scale of 0–4, where 0 is a normal pink healthy lung and 4 is a diffusely discolored dark red lung. **d** MERS-CoV lung titer 6 dpi in mice as described in **a**, **b**. Asterisks indicate statistical significance (*N* group described in **a** and **b**, *P* < 0.05) by one-way ANOVA with Kruskal–Wallis test for (**c**, **d**). Data for **a**–**d** are compiled from two independent experiments. For the box and whisker plots, the boxes encompass the 25th to 75th percentile, the line is at the median, while the whiskers represent the range. **e** Representative photomicrographs of MERS-CoV antigen (brown) and hematoxylin stained nuclei (blue) in mouse lung tissue sections from 6 dpi. The black bar is 100 µM.

cells in the septae and in alveolar air spaces, neutrophils in the air spaces, and proteinaceous debris in the air spaces organizing into hyaline membranes (Fig. 7a). Using the ATS ALI scoring tool described in Fig. 3, we found that only RDV therapy significantly (*P* = 0.005, one-way ANOVA with Kruskal–Wallis test) decreased lung injury scores (Fig. 7b). With a complementary histological tool, we then quantitated features of diffuse alveolar damage (DAD), the pathological hallmark of ALI[26]. We found that only therapeutic RDV reduced DAD scores (*P* = 0.04, one-way ANOVA with Kruskal–Wallis test) (Fig. 7c)[29]. Since degree of cell death appeared to correlate with protective efficacy, we quantitated levels of cleaved caspase-3 in lung tissue sections by antibody labeling. Caspase-3, a widely accepted marker of apoptosis, is a regulatory enzyme whose cleavage and activation drive programmed cell death[30]. Using the Definiens software suite, we obtained unbiased quantitative data showing that only RDV treatment significantly reduced levels of cleaved caspase-3 antigen (*P* = 0.0109, one-way ANOVA with Kruskal–Wallis test) (Fig. 7d). Thus, using three complementary and blinded approaches, we obtained similar data showing that only therapeutic RDV reduced histologic features of ALI.

## Discussion

Emerging viral diseases have caused significant global pandemics (e.g., HIV, 1918 influenza, smallpox), epidemics (e.g., SARS-CoV), and devastating outbreaks (e.g., EBOV, MERS-CoV). For CoV, metagenomic studies in wild animals have revealed a great diversity of viruses and hosts, and have even identified viruses similar to current and past epidemic strains in bats[31,32]. Thus, broad-spectrum therapies effective against known epidemic and zoonotic strains likely to seed future emergence, have the potential to diminish epidemic disease today and diminish future outbreaks. Currently, there are no FDA-approved treatments for any human CoV infection. Upon emergence of SARS- and MERS-CoV, patients were administered off-label antivirals (e.g., ribavirin, LPV, RTV) and immunomodulators (e.g., corticosteroids, interferon alpha-2a/2b, IFNb) as single agents or in combination in an attempt to ameliorate severe disease outcomes with very limited success[33]. Without randomized controlled trials, determining efficacy is difficult due to patient and treatment variability as well as a lack of appropriate matching controls. Although recent meta-analysis and modeling has suggested that interferon treatment does not improve clinical outcomes in MERS-CoV patients[4], the MIRACLE trial in the KSA is aimed at conclusively determining if a fixed dose combination of LPV/RTV-IFNb is effective at treating MERS-CoV infections[14]. Here, we show that RDV provides superior antiviral activity against MERS-CoV in vitro and in vivo as compared with LPV/RTV-IFNb. In addition, RDV was the only treatment to significantly reduce pulmonary pathology. Thus, we provide in vivo evidence of the potential for RDV to treat MERS-CoV infections. Efficacy

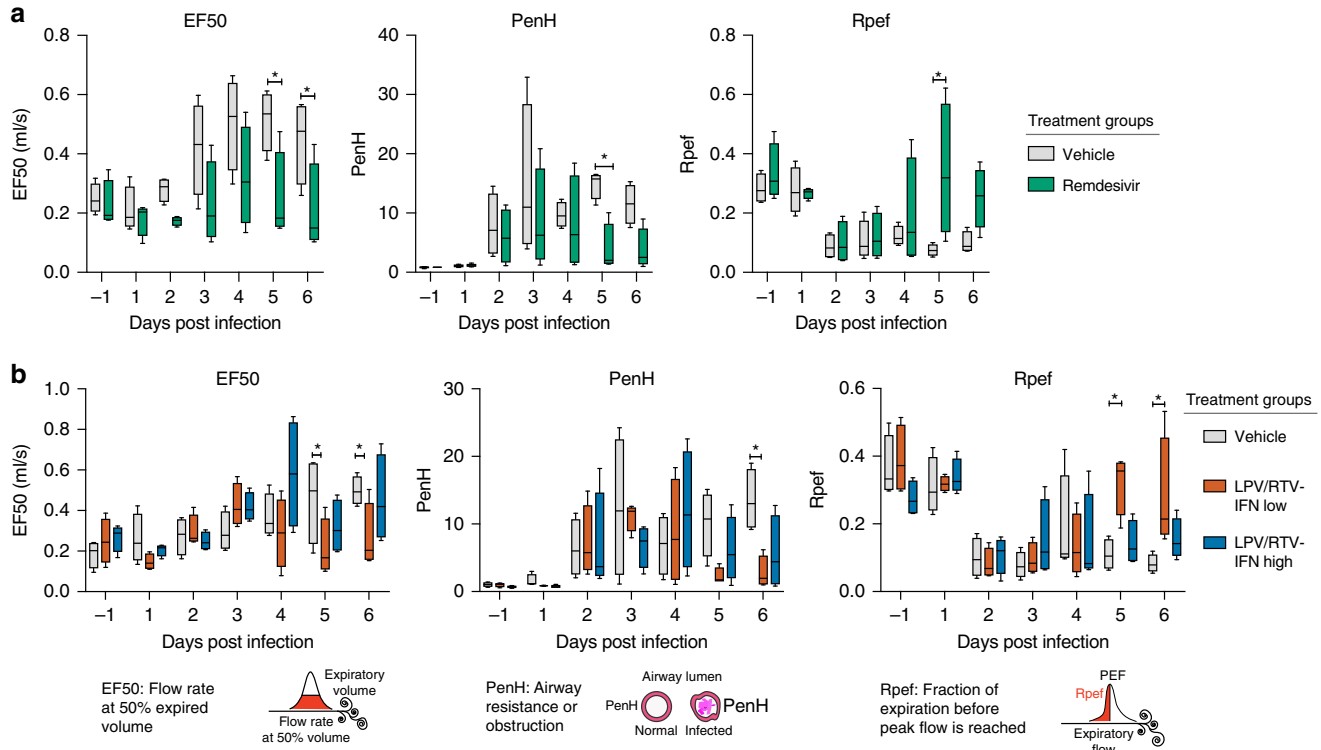

**Fig. 6 Therapeutic RDV and LPV/RTV-IFNb improve pulmonary function.** Whole-body plethysmography (WBP) was used to assess pulmonary function in mice. Representative WBP data for one of the two studies depicted in Fig. 5 are shown. All groups were $N = 4$ per day. EF50 is the flow rate at 50% expired volume. PenH is a surrogate measure of airway resistance. Rpef is the fraction of expiration before peak expiratory flow is reached. Altered EF50, PenH, and Rpef indicate bronchoconstriction or airway obstruction. Asterisks indicate statistical differences by two-way ANOVA with Sidek's multiple comparison test in **a** and two-way ANOVA with Tukey's multiple comparison test in **b**. The boxes encompass the 25th to 75th percentile, the line is at the median, while the whiskers represent the range.

testing of antiviral regimens in humans like those in the MIRA-CLE trial are essential to progress antiviral development and prioritize therapies most likely to improve clinical outcomes in MERS-CoV patients.

Differences in protein identity and concentration complicate the translation of in vitro antiviral activity to in vivo therapeutic efficacy. Although the $EC_{50}$ for LPV against MERS-CoV ($EC_{50} = 11\,\mu M$) falls within the maximum ($C_{max} = 15\,\mu M$) and minimum ($C_{min} = 9.5\,\mu M$) levels observed in human plasma, the biologically available fraction unbound to protein in these respective systems should be compared to more accurately translate in vitro activity to potential in vivo efficacy[9]. For example, the trough levels ($C_{min}$) of total (5.8 μM) and unbound protein free LPV (0.057 μM) in women with HIV differ by 100-fold, but LPV is still highly active and effective since the $EC_{50}$ (0.010–0.027 μM) falls below the unbound protein free $C_{min}$[7,34]. Together, these data argue that levels of free biologically active LPV achieved in humans are well below those that exert robust antiviral effects on MERS-CoV replication in cell culture systems. Thus, the micromolar $EC_{50}$ observed for LPV against MERS-CoV coupled with protein binding and insufficient levels of free LPV in plasma is likely responsible for the modest antiviral effect with LPV/RTV-IFNb prophylaxis and minimal impact on disease with therapeutic administration in our mouse model. In contrast, human equivalent doses of RDV are demonstrably efficacious in mice infected with SARS- and MERS-CoV and in nonhuman primates infected with EBOV thus demonstrating a more suitable PK/PD relationship[17].

Interferons are useful in treating multiple viral infections[35,36]. Since IFNb was shown to be the most potent against MERS-CoV when comparing the antiviral activity of multiple type I and type II interferons in Vero cells, IFNb was selected to use in the MIRACLE trial[8,10]. Mirroring the MIRACLE trial, we delivered IFNb subcutaneously every other day, which failed to reduce MERS-CoV viral loads and appeared to exacerbate disease in mice. As we sought to understand this result, we found little experimental congruency among reports detailing the prophylactic or therapeutic efficacy of type I interferons in animal models of MERS-CoV[11,37,38]. Falzarano et al. demonstrated that IFN-alpha-2a coupled with ribavirin initiated 8 h post infection improved outcomes in rhesus macaques and reduced viral copy number in lung tissue, but the treatment had no effect on infectious virus titers in bronchoalveolar lavage fluid[37]. In mice where hDPP4 is delivered to lung tissue by adenoviral transduction, intranasal IFNb given before or after MERS-CoV infection reduced lung titers although the peak lung titers in this model are approximately two orders of magnitude lower than the current transgenic models, and thus may be more easily treated[38–40]. The utility of the common marmoset as a model of MERS-CoV pathogenesis is controversial with one study detailing severe respiratory disease yet another reporting similarly mild disease among mock and MERS-CoV-infected animals[41,42]. In marmosets, Chan et al. explored the therapeutic potential of LPV/RTV or IFNb, but the small numbers of animals used per group, lack of time-matched viral load samples, and unexpected early mortality in the LPV group made the resultant data difficult to interpret[11]. Nevertheless, the studies noted above demonstrate that type I interferon can exert an antiviral effect on MERS-CoV in vivo when given subcutaneously (IFN alpha, rhesus macaque) and intranasally (IFNb, adenovirus hDPP4 model)[37–39]. Our inability to reduce MERS-CoV titer or improve outcomes with IFN as described above may be due to inherent differences in the animal

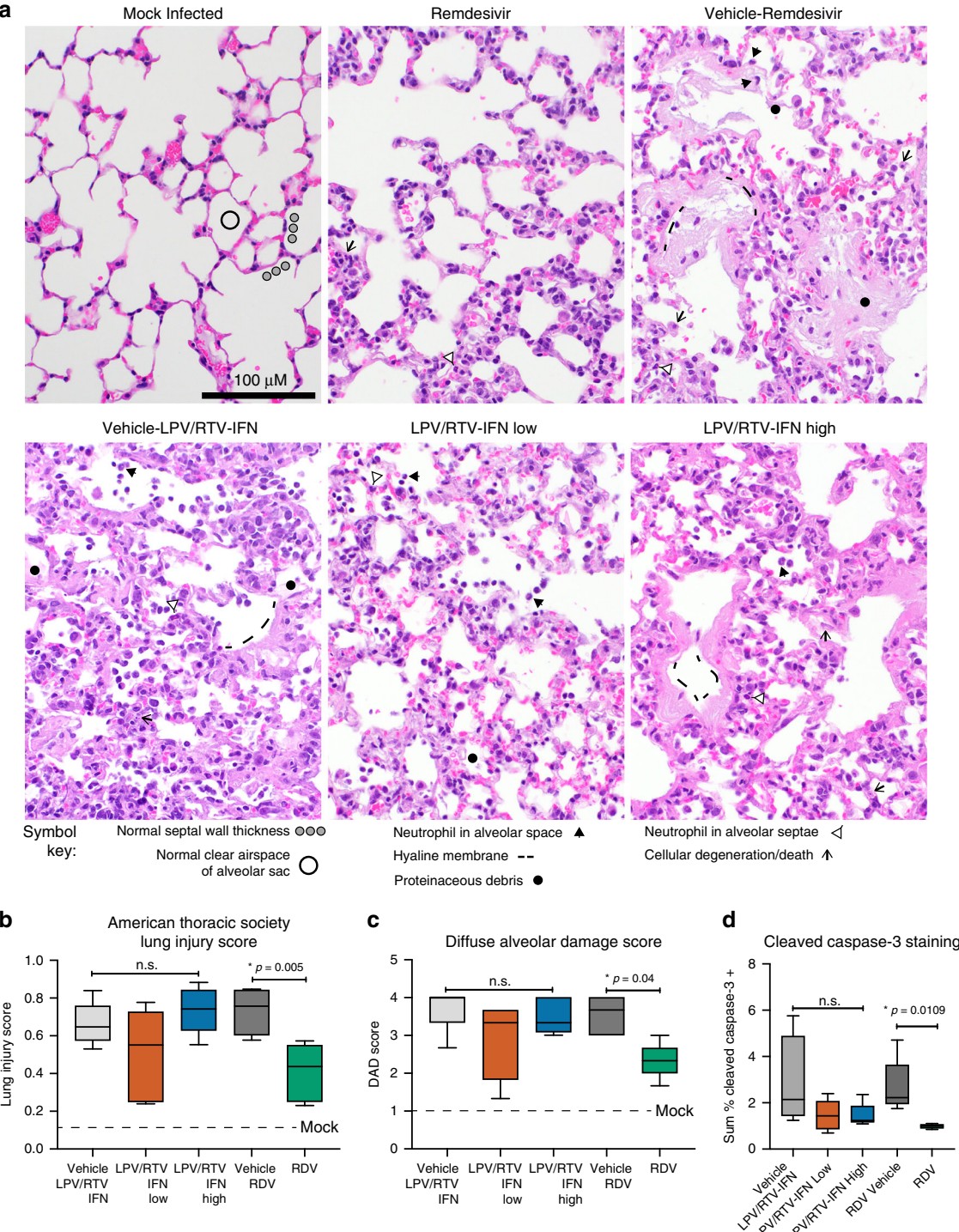

**Fig. 7 Therapeutic RDV but not LPV/RTV-IFNb diminishes signs of ALI. a** Representative images of the histological features of acute lung injury 6 dpi comparing a mock-infected mouse to the therapeutic treatment groups described in Figs. 5 and 6. Symbols identifying example features of disease are indicated in the figure. **b** American Thoracic Society Lung Injury Score derived as described in Fig. 3. The numbers of animals per group quantitated: vehicle RDV $N = 7$, RDV $N = 7$, vehicle LPV/RTV-IFNb $N = 9$, LPV/RTV-IFNb low $N = 7$, LPV/RTV-IFNb high $N = 8$. **c** Diffuse alveolar damage score quantitating the degree of cellular sloughing, necrosis, and breakdown of barrier epithelium and vascular leakage. For both **b** and **c**, scores were blindly assessed in three random high power (×60) fields of diseased lung tissue sections. **d** Quantitation of cleaved caspase-3 antigen staining in lung tissue sections from studies described in Figs. 5-7. Cleaved caspase-3 is a marker of cell death. The numbers of animals per group quantitated for all groups was $N = 5$/group. For the box and whisker plots, the boxes encompass the 25th to 75th percentile, the line is at the median, while the whiskers represent the range. For **b**–**d**, asterisks indicate statistical significance by one-way ANOVA and Kruskal–Wallis multiple comparison test.

models, delivery route, differences in IFN subtype and/or active viral antagonism of innate immunity. Since recent studies have demonstrated type III IFN to be most effective in ameliorating influenza pathogenesis in mice, comparative studies investigating the potency of different IFN subtypes should be pursued with MERS-CoV[43–45].

Acute lung injury (ALI) in humans is well defined by a set of clinical parameters (i.e., acute onset, diffuse bilateral infiltrates on X-ray, ratio of partial pressure of arterial oxygen to inspired oxygen < 300, no evidence of elevated pulmonary arterial pressure, etc.), which can be measured in mice but require specialized procedures, equipment, and training not readily available to most researchers[26,46]. Moreover, animal models of ALI typically fail to recapitulate all pathologic features observed in humans likely due to differences in underlying anatomy, physiology, immunology, genetics, and complex comorbidities typically associated with hospitalized ALI patients (e.g., diabetes, kidney and liver disease, etc.)[26]. For example, mice have a distinctly different lobar composition with less branching in the conducting airway than humans, a resting respiratory rate far exceeding that of humans (250–300 bpm in mice, 12–16 bpm in humans), and very different amounts of neutrophils in circulation (10–25% in mice, 50–70% in humans). To address these issues, the American Thoracic Society (ATS) has created tools to simplify the translation of mouse models of ALI to the human condition including those to quantitate the hallmarks of ALI in tissue sections[26]. We employed a complementary quantitative histologic assessment tool for diffuse alveolar damage (DAD), the pathological hallmark of ALI in humans[26,29]. With the ATS ALI and DAD histologic assessment tools described above, along with quantitation of cell death via cleaved caspase-3 antigen labeling, we show that only RDV therapy reduced ALI when initiated 1 dpi. Therefore, the evidence from several complementary histologic approaches demonstrate that RDV provides superior protection from ALI as compared with LPV/RTV-IFNb.

Remdesivir (RDV, GS-5734) is a broad-spectrum antiviral with potent in vitro efficacy against multiple genetically unrelated RNA viruses[16–18]. RDV has demonstrated in vivo efficacy against EBOV in nonhuman primates which has led to its inclusion in clinical studies evaluating the effects of RDV treatment in acute Ebola virus disease (EVD) as well as in EVD survivors with prolonged viral shedding[47,48]. As with our SARS-CoV studies with RDV, here we provide similar evidence for MERS-CoV with diminished weight loss, improved pulmonary function, and reduced virus replication with both prophylactic and therapeutic RDV[17]. The kinetics of SARS- or MERS-CoV replication and disease in mice is substantially accelerated as compared with that in humans. Virus replication in the lung peaks at 2 dpi in the lungs of mice infected with SARS-CoV or MERS-CoV and infection progresses to mortality or recovery by 7–10 dpi, depending on virus dose[17,24]. Therefore, the therapeutic window in which to treat mice prior to the peak of virus replication is only 1–2 days. In contrast, MERS-CoV replication in the human respiratory tract peaks at 7–10 days after the onset of symptoms and the disease course resolves or progresses to death within ~21 days[49,50]. Thus, the window for therapeutic administration after the onset of symptoms but prior to the peak of virus replication is very different in humans and experimentally infected mice. With therapeutic RDV, we observed decreased MERS-CoV pathogenesis and significant reductions in viral titers, yet therapeutic treatment did not completely abrogate disease. In addition, with high-titer virus inoculum, RDV was unable to prevent mortality and loss of pulmonary function although it did significantly reduce viral loads. These results are similar to those obtained for SARS-CoV, where therapeutic treatment initiated after peak virus titer and lung damage failed to improve outcomes

yet significantly reduced viral titer[17]. Similar observations were recently reported where neutralizing monoclonal antibodies failed to reduce severe lung disease pathology or clinical disease when administered 1 day after MERS-CoV infection in marmosets[51]. Since disease resulting from both SARS- and MERS-CoV infection is driven by both virus and host immune response factors, depending on the stage of the disease progression, early initiation of antiviral therapy, and/or holistic combination therapies will likely be needed to diminish virus replication, immunopathology, and/or promote repair and restoration of pulmonary homeostasis. Ongoing and future studies are aimed at determining if MERS-CoV is capable of generating resistance to RDV in vitro and whether the mutational spectra are similar to those obtained by MHV and SARS-CoV which shift $EC_{50}$ values only 3–5-fold[52].

In summary, we provide in vivo evidence of the potential utility of RDV in treating MERS-CoV patients. Overall, our work suggests that RDV may improve disease outcomes in CoV-infected patients, serve to protect health care workers in areas with endemic MERS-CoV and prove valuable in preventing future epidemics in the event of novel CoV emergence in the future.

## Methods

**Study design**. The primary goal of this study was to compare the prophylactic and therapeutic efficacy of RDV with the combination of LPV/RTV and IFNb. First, we assessed antiviral efficacy and cytoxicity in the Calu-3 human lung cell line as compared with the appropriate vehicle control. Experimental conditions in vitro were performed in sextuplicate unless otherwise stated, and antiviral assays were repeated four times per drug. Second, we evaluated the in vivo efficacy of prophylactic RDV as compared to vehicle with two different doses of MERS-CoV in a new transgenic mouse model of MERS-CoV pathogenesis with improved pharmacokinetics for nucleotide prodrugs. We performed two additional prophylactic studies comparing RDV and vehicle to LPV/RTV-IFNb, IFNb-alone, and their vehicles. Third, we assessed the therapeutic efficacy of the above treatment regimens in a mouse model of MERS-CoV pathogenesis, but did not include an IFNb, only arm and these studies were performed twice. All lung histological assessments were performed in a blinded manner. In addition, we performed a single therapeutic efficacy study with a lethal dose of MERS-CoV. Our in vivo efficacy studies were designed to mirror the ongoing MIRACLE human clinical trial which is evaluating combination LPV/RTV-IFNb. Our studies were intended to generate the data required to justify further testing in nonhuman primates and collectively inform future human clinical trials. Mice were age- and sex-matched and randomly assigned into groups before infection and treatment. Exclusion criteria for in vivo studies were as follows: If a given mouse unexpectedly did not lose weight after infection and their virus lung titers were more than 2 $\log_{10}$ lower than the mean of the group, this indicated that infection was inefficient, and all data related to that mouse were censored.

**Animal care and ethics statement**. Efficacy studies were performed in animal biosafety level 3 facilities at UNC Chapel Hill. All works were conducted under protocols approved by the Institutional Animal Care and Use Committee at UNC Chapel Hill (IACUC protocol #16-284) according to guidelines set by the Association for the Assessment and Accreditation of Laboratory Animal Care and the U.S. Department of Agriculture.

**Virus**. For in vitro studies, a MERS-CoV reporter virus expressing nanoluciferase was employed (MERS-nLUC)[17]. MERS-nLUC stocks were derived from a molecular clone through electroporation of Vero 81 cells (ATCC CCL-81) and isolation of virus through harvesting of culture supernatants[53]. Briefly, DNA fragments A–F encoding MERS-nLUC cDNA genome were ligated to create full-length cDNA, which was then used as a template for in vitro transcription. Full-length genomic RNA was then electroporated into Vero-81 cells yielding recombinant virus stock. The resultant stock was passaged twice in Vero 81 to generate a working stock (1.6E + 07 pfu/mL) for our studies. Wild-type MERS-CoV for comparative antiviral efficacy studies was derived from our EMC 2012 infectious clone as described above to obtain a working stock with a titer of 3E + 07 pfu/mL[53]. For in vivo studies, we utilized mouse adapted MERS-CoV passage 35 clone 4 (MERS M35C4)[24]. MERS M35C4 has 12 amino acid coding changes as well as a single-nucleotide change in the 5′ UTR and a large deletion in ORF4b/ORF5. This virus is a clonal isolate generated through serial passage of MERS-CoV in mice. After 35 passages in mice, virus was plaque purified and clone 4 was expanded two times on Vero CCL81 cells to obtain our working stock. The working stock (1.1E + 08 pfu/mL) was created in virus collection medium (Optimem (Gibco), 3% Fetal Clone II serum product (Hyclone), and antibiotic/antimycotic (Gibco) and non-essential amino acids (Gibco)).

**Compounds and formulation for in vitro studies**. Remdesivir (RDV), lopinavir (LPV), and ritonavir (RTV) were solubilized in 100% DMSO and provided by Gilead Sciences, Inc. Recombinant human interferon beta (IFNb) protein was purchased from R&D Systems (8499IF010/CF, $2.8 \times 10^8$ IU/mg compared with WHO standard) and solubilized in sterile water as recommended.

**In vitro efficacy and cytotoxicity in Calu-3 cells**. To determine if virus replication kinetics and RDV susceptibility were similar among wild-type (WT) MERS-CoV EMC 2012 and MERS-nLUC, we performed comparative antiviral assays the human lung epithelial cell line, Calu-3 2B4 (kindly provided by Dr. Chien Tseng University of Texas Medical Branch). Calu-3 2B4 was maintained in the DMEM (Gibco), 20% fetal bovine serum (FBS, Hyclone), and 1× Antibiotic–Antimycotic (A/A, Gibco). Briefly, 48 h prior to infection, Calu-3 cells were plated at 4.3E + 04 cells/well. Twenty-four hours prior to infection, culture medium was exchanged with fresh medium. Calu-3 2B4 cells were infected with WT EMC 2012 or MERS-nLUC at a multiplicity of infection (MOI) of 0.1 for 1 h at 37 °C after which cells were washed, and a dose response of RDV diluted in twofold steps in media (DMEM, 10% FBS, DMEM, 1x A/A) was added in duplicate. Cells were then incubated at 37 °C in 5% $CO_2$ for 24 h after which 100 µl of media from each well was collected and assayed for virus production by plaque assay in Vero CCL81 cells. Briefly, 500,000 Vero CCL81 cells/well were seeded in six-well plates. The following day, medium was removed, and serial dilutions of sample were added per plate ($10^{-1}$–$10^{-6}$ dilutions) and incubated at 37 °C for 1 h, after which wells were overlayed with 1× DMEM, 5% Fetal Clone 2 serum, 1× A/A, 0.8% agarose. Three days after, plaques were enumerated to generate a plaque/ml value. The $IC_{50}$ value was defined in GraphPad Prism 7 (GraphPad).

To better understand the antiviral activities of RDV, LPV, RTV, IFNb, or combinations of LPV/RTV or LPV/RTV/IFNb, we performed efficacy and cytotoxicity assays in human lung epithelial cells (Calu-3). Cells were plated as described above, and infected with MERS-CoV expressing nanoluciferase (MERS-nLuc) in sextuplicate at an MOI of 0.08 for 1 h in the presence of a dose response of drug as described below. After 1 h of infection, virus was removed, cultures were rinsed once with medium, and fresh medium was added containing dilutions of drug. RDV (i.e., GS-5734, stock at 20 mM) was serially diluted in 100% DMSO in twofold increments to obtain a ten-point dilution series. Human IFNb (R + D Systems, 200 µg/mL or $5.6 \times 10^7$ IU/mL) was similarly diluted in PBS. LPV (10 mM stock) and RTV (10 mM stock) were similarly diluted in DMSO, although various amounts of stock compound were added directly to media to obtain the top four dilutions in the ten-point curve (50 µM, 25 µM, 12.5 µM, 6.25 µM). To model the antiviral effect of the fixed dose combination of LPV/RTV used to treat HIV, we performed antiviral assays with a fixed combination of LPV/RTV (weight:weight ratio of 4:1 or molar ratio 4.6:1). The LPV/RTV combination was serially diluted in 100% DMSO in twofold increments, but mixture stock was added directly to the media similar to above to obtain the top four concentrations. These studies were repeated in 4 independent experiments.

To model LPV/RTV in combination with IFNb, we first performed comparative equilibrium dialysis (CED)[23] and determined that 5 µM LPV in 10% FBS containing cell culture medium gave the same amount of free (i.e., unbound to protein) LPV as the maximum concentration attained in human plasma ($C_{max}$, i.e., 15 µM). We then combined a dose response of human IFNb with a fixed concentration of LPV/RTV (molar ratio of 4.6:1, (5 µM Lopinavir and 1.1 µM Ritonavir) based on the human plasma equivalent maximal concentration of LPV as determined by CED (i.e., 5 µM). Thus, this study was aimed at determining if the maximal amount of LPV attainable in humans provided an additive or synergistic antiviral effect when combined with IFNb. All studies were performed in cell culture medium containing 10% FBS. DMSO (0.5%) was constant in all conditions. At 48 h post infection (hpi), virus replication was quantified on a Spectramax (Molecular Devices) via nanoluciferase assay (NanoGlo Promega). Values from replicate wells per condition were averaged and compared with controls to generate a percent inhibition value for each drug dilution. The $IC_{50}$ value was defined in GraphPad Prism 7 (GraphPad) as the concentration at which there was a 50% decrease in viral replication using UV-treated MERS-nLUC (100% inhibition) and vehicle alone (0% inhibition) as controls. To measure cytotoxicity, cells were exposed to the same drug dilutions and controls as the efficacy studies, but in the absence of virus infection. After 48 h of exposure, cell viability was determined by Cell-Titer-Glo Assay (Promega) and quantitated on a Spectramax. Similar data was obtained in at least three independent experiments.

**Formulations for in vivo studies**. RDV was solubilized at 2.5 mg/ml in vehicle containing 12% sulfobutylether-β-cyclodextrin sodium salt in water (with HCl/NaOH) at pH 5.0. LPV (32 mg/mL) RTV (8 mg/mL) was solubilized in vehicle containing 90% propylene glycol and 10% ethanol. The ratio of LPV to RTV was held at 4:1 (weight:weight) for all studies herein. Recombinant mouse IFNb protein was purchased from R&D Systems (8234-MB/CF, $1.2 \times 10^9$ IU/mg calibrated against Murine IFN-beta WHO International Standard) for the in vivo studies and reconstituted in PBS.

**Animals**. MERS-CoV binds the human receptor dipeptidyl peptidase 4 (DPP4) to gain entry into cells, and two residues (288 and 330) in the binding interface of the mouse ortholog prevent infection of mice. We recently developed a mouse model for MERS-CoV through the mutation of mouse DPP4 at 288 and 330 via CRISPR/Cas9 thus humanizing the receptor ($hDPP4$) and rendering mice susceptible to MERS-CoV infection[22]. The serum carboxylesterase 1c (Ces1c) in mice drastically reduces RDV stability. Thus, all in vivo studies must be performed in mice genetically deleted for Ces1c (*Ces1c*$^{-/-}$, stock 014096, The Jackson Laboratory)[17]. In order to perform in vivo efficacy studies with RDV and MERS-CoV in mice, we generated humanized DPP4 mice deficient in Ces1c expression (C57BL/6J *Ces1c*$^{-/-}$ *hDPP4*, The Jackson Laboratory stock number 403188). Briefly, our C57BL/6J *hDPP4* mouse was rederived using *Ces1c*$^{-/-}$ oocytes. The resultant heterozygous mice were crossed with *Ces1c*$^{-/-}$ mice to generate mice homozygous for the Ces1c deletion (*Ces1c*$^{-/-}$) and heterozygous for the *hDPP4* alleles. The resultant offspring were genotyped and bred to generate founders homozygous for both *Ces1c*$^{-/-}$ and *hDPP4*.

**MERS-CoV pathogenesis in *Ces1c*$^{-/-}$ and *hDPP4* mice**. To determine if the pathogenesis of MERS-CoV in *Ces1c*$^{-/-}$ and *hDPP4* was similar to the parental *hDPP4* line, we performed pathogenesis studies in the newly created *Ces1c*$^{-/-}$ and *hDPP4* line. Similar numbers ($N = 9$–10/sex/group) of 23–24-week-old male and female mice were randomly assigned to each infection group. Mice were anaesthetized with a mixture of ketamine/xylazine and then intranasally infected with either 5E + 04 or 5E + 05 pfu MERS M35C4 in 50 µl virus collection medium. To monitor morbidity, mice were weighed daily out to 6 dpi. Mice that lost >20% of their starting weight were humanely sacrificed by isofluorane overdose. To better understand the magnitude of MERS-CoV replication and the typical metrics of disease, we infected 12 9–12-week-old female mice with 5E + 04 MERS M35C4 in 50 µl virus collection medium as done above. To monitor morbidity, mice were weighed daily. A subset of each cohort was randomly assigned for pulmonary function measurements by whole-body plethysmography (WBP, Data Sciences International) daily[28]. On 6 dpi, animals were killed by isoflurane overdose, lungs were scored for lung hemorrhage, and the inferior right lobe was frozen at −80 °C for viral titration via plaque assay as described above[22]. Lung hemorrhage is a gross pathological phenotype readily observed by the naked eye driven by the degree of virus replication where the coloration of the lung changes from pink to dark red[25,26]. The large left lobe was placed in 10% buffered formalin and stored at 4 °C for 1–3 weeks, until histological sectioning and analysis. Lung sectioning, hematoxylin and eosin staining as well as MERS-CoV antigen (primary antibody 1:500, sera from mice vaccinated with MERS-CoV nucleocapsid antigen) staining was performed by the Animal Histopathology & Laboratory Medicine Core at UNC.

**Interferon beta pharmacodynamic studies**. To generate a pharmacokinetic and pharmacodynamic (PK/PD) relationship for IFNb, we subcutaneously administered mouse IFNb (R + D Systems) to 18–20 week-old male and female *Ces1c*$^{-/-}$ or *Ces1c*$^{-/-}$ *hDPP4* mice. Human equivalent dosing in mice was calculated based on the recommended dosing strategy for Betaseron (human IFNb, 0.13 million international units (MIU)/kg every other day)[54]. The human dose was multiplied by 12.3 in order to get the mouse equivalent dose of 1.6 MIU/kg or 4.8E4 IU/30 g mouse (40 µg/30 g mouse) based on body surface area[55]. First, we compared 1× and 2.5× human equivalent doses of IFNb in cohorts ($N = 20$/group) of *Ces1c*$^{-/-}$ mice. At 2, 4, 8, and 12 h post treatment, five mice per group were humanely euthanized, and plasma was snap-frozen at −80 °C, and peripheral blood mononuclear cells (PBMC) were isolated, place in Trizol LS (Thermo Fisher) and stored at −80 °C until analysis. Second, we compared the responses to a 25× human equivalent dose of IFNb (i.e., 40 MIU/kg or 1.2 MIU/30 g mouse (1000 µg/30 g mouse)) in *Ces1c*$^{-/-}$ or *Ces1c*$^{-/-}$ *hDPP4* mice ($N = 25$/group) to ensure that these mouse lines respond similarly. At 0, 2, 4, 8, and 12 h post treatment, mice were humanely euthanized and plasma and PBMCs were isolated and stored as noted above. In addition, lungs were isolated at 4 and 8 h post treatment and stored in RNAlater (Thermo Fisher) at −80 °C until analysis. The total RNA was isolated from PBMCs using Zymo Research Direct-Zol RNA mini kit. Lung tissue was homogenized in Trizol (Thermo Fisher), and the total RNA was isolated similarly to PBMCs. The response to IFNb treatment in PBMCs was quantitated by qRT-PCR for the classic interferon stimulated gene MX dynamin-like GTPase 1 (*Mx1*) by TaqMan assay (Thermo Fisher Mm00487796_m1) using the Taqman Fast Virus 1-Step Master Mix (Thermo Fisher). *Mx1* gene expression was compared with signals from the housekeeping gene glyceraldehyde-3-phosphate dehydrogenase (*GAPDH*, Thermo Fisher, Mm99999915_g1) and fold change over mock or time 0 post treatment was calculated by the ΔΔCt method[56]. To monitor induction of interferon induced protein expression, we quantitated levels of mouse IFN-gamma inducible protein 10 (IP10, CXCL-10) in plasma via ELISA (Invitrogen).

**Prophylactic in vivo efficacy studies**. For all drug studies, mice that lost >20% of their starting weight were weighed twice and subjected to an additional visual check each day for clinical signs (hunching, ease of mobility, lethargy, etc.). Mice that fell <70% of their starting weight were immediately euthanized. Mortality was defined as unexpectedly finding mice dead in the cage.

Several prophylactic studies were performed to determine if drug regimens could affect virus replication and/or disease progression. In the initial study, we only compared RDV and vehicle. Briefly, groups of 9–12-week-old male and female *Ces1c*$^{-/-}$ *hDPP4* were randomly assigned to groups ($n = 7$–9) and acclimated for

5–7 days at biosafety level 3 (BSL3). Treatment with vehicle (see above) or RDV (25 mg/kg, BID subcutaneously) was initiated the day prior to infection. For MERS-CoV infection, mice were anaesthetized with a mixture of ketamine/xylazine and then intranasally infected with either 5E + 04 or 5E + 05 pfu MERS M35C4 in 50 μl virus collection medium. To monitor morbidity, mice were weighed daily. On 4 or 6 dpi, animals were killed by isofluorane overdose, lungs were scored for hemorrhage (described above), and the inferior right lobe was frozen at −80 °C for viral titration via plaque assay as described above[22]. The large left lobe was placed in 10% buffered formalin and stored at 4 °C for 1–3 weeks, until sectioning and histological analysis. Lung sectioning, hematoxylin and eosin staining, as well as MERS-CoV antigen staining as described above were performed by the Animal Histopathology & Laboratory Medicine Core at UNC. Due to increased variability of infection and outcomes in male mice, all following studies were performed in only female mice.

We then performed two prophylactic studies to ascertain whether LPV/RTV-IFNb or IFNb alone could affect virus replication or disease progression. The studies were designed to mirror therapeutic efficacy studies. In the first study, in randomly assigned groups (n = 9–10) of 11–13-week-old Ces1c−/− hDPP4 female mice, we compared vehicle, LPV/RTV-IFNb, or IFNb alone. All treatments were initiated one day prior to infection. For MERS-CoV infection, mice were anaesthetized with a mixture of ketamine/xylazine and then intranasally infected with either 5E + 04 pfu MERS M35C4 in 50 μl virus collection medium. A human equivalent dose of a coformulation of LPV (160 mg/kg) and RTV (40 mg/kg) at 5 mL/kg was administered once daily via oral gavage. Groups receiving a 1× human equivalent dose of IFNb (R + D Systems, 1.6 MIU/kg or 4.8E4 IU/30 g mouse (40 μg/30 g mouse)) were dosed every other day via subcutaneous injection. To control for potential vehicle effects in the LPV/RTV-IFNb groups, we administered oral vehicle (90% propylene glycol and 10% ethanol) daily and subcutaneously injected with PBS every other day. RDV (25 mg/kg) at 10 mL/kg was administered twice daily via subcutaneous injection. As a control for RDV, an additional group was given subcutaneous vehicle. Mice were weighed daily to monitor morbidity. A subset (n = 4) of each cohort was randomly assigned for pulmonary function measurements by whole-body plethysmography (WBP, Data Sciences International) daily[28]. On 2 dpi, three animals per group were killed by isofluorane overdose, lungs were scored for hemorrhage (described above), and the large left lobe was frozen at −80 °C for viral titration via plaque assay as described above[22]. On 6 dpi, animals were killed and processed as done on 2 dpi.

The second prophylactic study was designed to optimize the potential effects IFNb based on the PK/PD studies described above. Given that interferon stimulated gene expression peaks between 2 and 4 h post administration, in this study all groups receiving IFNb treatments were started on treatment 2 h prior to infection and every other day thereafter. Moreover, to maximize the potential IFN effect in the IFNb only group, we utilized a 25× human equivalent dose rather than utilize a 1× human equivalent dose as done in the previous prophylactic study. In randomly assigned groups of 12–14-week-old Ces1c−/− hDPP4 female mice, we compared vehicle, LPV/RTV-IFNb low (1.6 MIU/kg or 4.8E4 IU/30 g mouse (40 μg/30 g mouse)), LPV/RTV-IFNb high (40 MIU/kg or 1.2 MIU/30 g mouse (1000 μg/30 g mouse)), and IFNb high alone. The LPV/RTV dose amounts and schedule were similar to that in the previous study. To control for dosing effects, vehicle-treated mice received both oral vehicle and subcutaneous PBS to mirror IFNb injections. Likewise, IFNb only group received oral vehicle to mirror that seen in orally dosed groups. Similar to the previous study, RDV and its vehicle were administered as a control. MERS-CoV infection was performed exactly as described in the previous study. Mice were weighed daily to monitor morbidity. Unlike previous studies, pulmonary function measurements by WBP (Data Sciences International) were performed on all mice in all groups until 2 dpi, after which these measurements were performed on all remaining mice per group (n = 6)[28]. On 2 dpi, three animals per group were killed by isofluorane overdose, lungs were scored for hemorrhage (described above), and the large left lobe was frozen at −80 °C for viral titration via plaque assay[22]. On 6 dpi, animals were killed and processed as done on 2 dpi.

**Therapeutic in vivo efficacy studies**. For head-to-head therapeutic efficacy studies comparing RDV to LPV/RTV-IFNb, female 9–12-week-old Ces1c−/− hDPP4 were randomly assigned to each treatment group (n = 10–12). After a 5–7 day acclimation time at BSL3, mice were anaesthetized with a mixture of ketamine/xylazine and then intranasally infected with 5E + 04 pfu of MERS M35C4 in 50 μl virus collection medium (see above). One day post infection, treatment was initiated. For LPV/RTV groups, mice were administered a human equivalent dose of a coformulation of LPV (160 mg/kg) and RTV (40 mg/kg) at 5 mL/kg once daily via oral gavage. Animals that received LPV/RTV also received mouse IFNb (R + D Systems) every other day at one of two doses via subcutaneous injection. The high IFNb dose group was administered a 25× human equivalent dose of 40 MIU/kg or 1.2 MIU/30 g mouse (1000 μg/30 g mouse). The low IFNb dose group was administered as 1× human equivalent dose of 1.6 MIU/kg or 4.8E4 IU/30 g mouse (40 μg/30 g mouse). To control for potential vehicle effects in the LPV/RTV-IFNb groups, we administered oral vehicle (propylene glycol, ethanol) daily and subcutaneously injected with PBS every other day. RDV (25 mg/kg) at 10 mL/kg was administered twice daily via subcutaneous injection. As a control for RDV, an additional group was given subcutaneous vehicle as a control. To monitor morbidity, mice were weighed daily. A subset of each cohort was randomly assigned for pulmonary function measurements by WBP (Data Sciences

International) daily[28]. On 6 dpi, animals were killed by isoflurane overdose, lungs were scored for hemorrhage (described above), and the large left lobe was frozen at −80 °C for viral titration via plaque assay as described above[22]. The inferior right lobe was placed in 10% buffered formalin and stored at 4 °C for 1–3 weeks, until sectioning and histological analysis. Lung sectioning, hematoxylin and eosin staining, as well as MERS-CoV antigen staining as described above were performed by the Animal Histopathology & Laboratory Medicine Core at UNC.

**Acute lung injury histological assessment tools**. We used two different and complementary quantitative histologic tools to determine if antiviral treatments diminished the histopathologic features associated with lung injury. Both analyses and scoring were performed by a Board Certified Veterinary Pathologist who was blinded to the treatment groups.

The first tool is a Lung Injury Scoring System that was created by the American Thoracic Society in order to help quantitate histological features of ALI observed in mouse models and increase their translation to the human condition[26]. In a blinded manner, we chose three random diseased fields of lung tissue at high power (60 ×), which were scored for the following: (A) neutrophils in the alveolar space (none = 0, 1–5 cells = 1, > 5 cells = 2), (B) neutrophils in the interstitial space/septae (none = 0, 1–5 cells = 1, > 5 cells = 2), (C) hyaline membranes (none = 0, one membrane = 1, > 1 membrane = 2), (D) Proteinaceous debris in air spaces (none = 0, one instance = 1, > 1 instance = 2), (E) alveolar septal thickening (< 2× mock thickness = 0, 2–4× mock thickness = 1, > 4× mock thickness = 2). To obtain a lung injury score per field, the scores for A–E were then put into the following formula, which contains multipliers that assign varying levels of importance for each phenotype of the disease state.: score = [(20x A) + (14 x B) + (7 x C) + (7 x D) + (2 x E)]/100. The scores for the three fields per mouse were averaged to obtain a final score ranging from 0 to and including 1.

The second histological tool to quantitate lung injury was reported by Schmidt et al., where they used this tool to quantitate diffuse alveolar damage (DAD) in mice infected with RSV[29]. DAD is the pathological hallmark of ALI[26,29,46]. Similar to the implementation of the ATS tool described above, we scored three random diseased fields of lung tissue at high power (60 ×) for the following in a blinded manner: 1 = absence of cellular sloughing and necrosis, 2 = Uncommon solitary cell sloughing and necrosis (1–2 foci/field), 3 = multifocal (3 + foci) cellular sloughing and necrosis with uncommon septal wall hyalinization, or 4 = multifocal ( >75% of field) cellular sloughing and necrosis with common and/or prominent hyaline membranes. The scores for the three fields per mouse were averaged to get a final DAD score per mouse.

**Cleaved caspase-3 antigen quantitation**. The active and cleaved form of caspase-3 can be differentiated by immunohistochemistry. Lung tissue sections were stained for cleaved caspase-3 antigen (1:500, Cell Signaling #9664) by the Animal Histo-pathology & Laboratory Medicine Core at UNC. At the UNC Translational Pathology Laboratory (TPL) Core, stained slides were scanned on an Aperio ScanScope XT (Leica Biosystems) with a 20X objective and a camera resolution of 0.4942 microns per pixel. Images were analyzed in Definiens Architect XD 2.7 with Tissue Studio version 4.4.2. Within each tissue section, total area was calculated. Single cells were detected initially based on hematoxylin staining of nuclei, and cell borders were interpolated in Tissue Studio. Each cell was then scored for cleaved caspase staining intensity based on a linear scale of Low, Medium, and High scale. The sum of the percent positive for high and medium caspase-3 staining was graphed for our studies. Tissue sections of diseased mouse intestine was used as a positive control.

**Statistical analysis**. All statistical data analyses were performed in Graphpad Prism 7. Statistical significance for each endpoint was determined with specific statistical tests. For each test, a P-value < 0.05 was considered significant. Specific tests to determine statistical significance are noted in each sure legend.

**Reporting summary**. Further information on research design is available in the Nature Research Reporting Summary linked to this article.

## Data availability
Primary data for Figs. 1–7, and Supplementary Figs. 1–6 are provided in the Source Data file.

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

## Acknowledgements

We would like to thank Dr. Daphne Ma for superb organization of mouse breeding efforts. Animal histopathology and/or clinical services was performed by the Animal Histopathology & Laboratory Medicine Core at the University of North Carolina, which is supported in part by an NCI Center Core Support Grant (5P30CA016086-41) to the UNC Lineberger Comprehensive Cancer Center. We thank Bentley Midkiff in the UNC Translational Pathology Laboratory (TPL) for expert technical assistance. The UNC TPL is supported in part, by grants from the NCI (5P30CA016086-42), NIH (U54-CA156733), NIEHS (5 P30 ES010126-17), UCRF, and NCBT (2015-IDG-1007). We would like to

acknowledge the following funding sources, Antiviral Drug Discovery and Development Center (5U19AI109680), a partnership grant from the National Institutes of Health (5R01AI132178), and an NIAID R01 grant (AI108197). A.C.S. received a contract from Gilead Sciences to support the in vitro and in vivo efficacy studies reported herein.

## Author contributions

A.C.S. and J.Y.F. designed in vitro efficacy studies. A.C.S., J.Y.F., and T.P.S. executed and/or analyzed in vitro efficacy studies. T.P.S., D.B., A.H., R.J., T.C., and R.S.B. designed in vivo efficacy studies. T.P.S., A.J.B, J.W., A.S., and S.R.L. executed and analyzed in vivo efficacy studies. A.S. and S.R.L. performed whole-body plethysmography for in vivo studies. S.A.M. assessed all lung pathology. M.O.C., J.E.S., L.B., and S.S. were responsible for synthesis, scale-up and formulation of small molecules. T.P.S., A.C.S., J.Y.F., R.J., A.H., D.P., T.C., M.D., and R.S.B. wrote the paper.

## Competing interests

A.C.S. received a contract from Gilead Sciences to support the in vitro and in vivo efficacy studies reported herein. These authors are employees of Gilead Sciences and hold stock in Gilead Sciences: Alison Hogg, Darius Babusis, Michael O. Clarke, Jamie E. Spahn, Laura Bauer, Scott Sellers, Danielle Porter, Joy Y. Feng, Tomas Cihlar, and Robert Jordan.
