## [Peer Review File · Nature Communications]

Reviewers' Comments:

Reviewer #1:

Remarks to the Author:

This manuscript by Sheahan et al. describes both in vitro and in vivo comparisons of the efficacy of therapeutic treatments against Middle East Respiratory Syndrome Coronavirus (MERS-CoV). Coronaviruses can emerge into the human population to cause pandemic disease. Currently there are no FDA approved drugs to treat CoV infections in humans, therefore the demonstration of therapeutic efficacy for an antiviral therapy for an emerging coronavirus infection is highly significant. Previously, Sheahan and co-workers showed that nucleotide prodrug remdesivir (GS-5734) is effective in reducing replication and respiratory symptoms in mice infected with Severe Acute Respiratory Syndrome (Sheahan et al., Science Translational Medicine, 2017). In that paper, they describe how mice lacking a serum esterase that reduces the stability of prodrugs like remdesivir, termed *Ces1c*^{-/-} mice, are better models for evaluating the therapeutic efficacy of these antivirals. In this study, Sheahan and co-workers first compared the efficacy of remdesivir versus interferon beta or a combination treatment of HIV protease inhibitors with interferon beta treatment for the ability to inhibit virus replication in cell culture. They found that the EC50 is much lower for remdesivir (0.09 μ M) compared to any other inhibitor tested (Figure 1). They then evaluated the efficacy of the treatments using the *Ces1c*^{-/-}-hDPP4 mice (expressing the receptor for MERS-CoV and having the longer half-life of the prodrug) and report that remdesivir reduces MERS-CoV disease (Fig 2). They show that therapeutic remdesivir diminishes MERS-CoV disease (Fig 3) whereas other treatments are not effective (Fig 4). Importantly, the authors describe the strengths of their animal model system and the limitations of the use of lopinavir and ritonavir in treating coronavirus infections. The discussion section is excellent as it provides extensive comparisons to other published studies that use animal model systems. Results are well described, the model system is excellent, both male and female mice were used for the study. Supplemental information is also well done. Overall, this outstanding study provides a comprehensive in vitro (Calu3 cells) and in vivo (*Ces1c*^{-/-}-hDPP4 mice) efficacy data describing superior therapeutic efficacy of remdesivir (GS-5734) compared to other treatments.

Minor comments for the authors' consideration:

1. Abstract and Intro: Consider revising the sentences with KSA in them. MERS-CoV respiratory disease spread worldwide, but was first recognized in patients from the Kingdom of Saudi Arabia.
2. Abstract: consider revising the abstract to better describe the mouse model system. Using a mouse model deleted for an esterase that limits prodrug stability and humanized for the receptor (*Ces1c*^{-/-}-hDPP4 mice), we...
3. Introduction, line 12: replace locally with in the Middle East
4. Introduction, last sentence: revise to better describe the animal model. The deletion of the esterase is not new (Sheahan et al., 2017); the new part is that the mouse model has the humanized receptor (Cockrell et al., 2016) and is deleted for the esterase, which allows for the first time the evaluation of these antivirals against MERS-CoV infections in vivo.
5. Discussion should include some comment about future studies evaluating the emergence of coronavirus drug resistant mutants. Is it likely the same or different for SARS-CoV versus MERS-CoV?

Reviewer #2:

Remarks to the Author:

This study demonstrates in vivo efficacy of an antiviral compound, RDV, previously shown to be active against various coronaviruses, in a transgenic mouse model using a mouse-adapted MERS-CoV. In general, the experiments are well performed, and efficacy of RDV is evident at least in pretreatment and in a more therapeutic approach when applied very early after initial infection.

Major:

- Viruses designed to express reporter genes are usually highly attenuated. As such, it is not clear how effective RDV would be against the wt virus (as compared to MERS-nLuc). Similarly, the authors use a mouse adapted MERS-CoV in their study, without providing data on the amount and location of adaptive mutations in their virus. It remains unclear whether RDV would be similarly active towards a (human) MERS-CoV isolate.

- Histologies. Fig 2 E/G: It is impossible to define an Alveolar type I or type II cell on these sections "based on morphology". If the authors want to make this claim, cell-specific markers for these cells need to be applied together with the MERS-Cov Ab.

It is also not possible to differentiate edema and hyaline membranes at this resolution. In Fig 3 (results text), what is meant by "markers of ARDS" beyond hyaline membranes? Generally, in all the histologies in Fig 2-4, it is surprising that there are few alveolar infiltrates in MERS-CoV infected mice, a hallmark of respiratory virus-induced ARDS. This is particularly low in Fig 3F. I am not sure whether the inflammatory pathology the authors observe in their MERS-CoV model reflects ARDS at all, as the standard criteria are not obvious and it is hard to see from the small sections at the given resolution. Have the sections be analysed by a lung pathologist? Also, the body measurements rather point at an obstructive airway disease, and lack the ARDS-typical reduction of vital capacity and compliance.

- Did therapeutically applied RDV affect survival, as a major translational finding that would allow to move into a clinical trial?

- How does a pretreatment with LPV/r (plus and minus IFN) affect titers/outcomes in the mouse model? Only therapeutic treatment data are provided. In this line, the discussion on IFN-beta as a main "driver" of improved outcome in the LPV/r + IFN is in fact not supported by data. LPV/r versus LPV/r + IFN has not been tested in vivo to make this claim. Moreover, there is a lot of evidence in the literature that IFN type I when applied in the course of respiratory viral infection rather drives immunopathology (including lack of barrier function/hemorrhage) and contributes to morbidity in vivo (eg, PMID 27520969, PMID 28514692), which could explain the data obtained in the LPV/r + IFN groups. On a minor note, the ARDS trial mentioned here by the authors (ref 32) using type I IFN in ARDS did not prove efficacy).

Minor

- The the authors should provide information on the detailed antiviral mechanism of action of RDV in the introduction.

Response to Reviewers' Comments:

Reviewer #1 (Remarks to the Author):

Minor comments for the authors' consideration:

1. Abstract and Intro: Consider revising the sentences with KSA in them. MERS-CoV respiratory disease spread worldwide, but was first recognized in patients from the Kingdom of Saudi Arabia.

Response: We thank the review for this comment. We have since eliminated the text pointed out by the reviewer in the abstract and have ensured that the introduction describes MERS-CoV as a disease that is not necessarily confined to KSA's borders.

2. Abstract: consider revising the abstract to better describe the mouse model system. Using a mouse model deleted for an esterase that limits prodrug stability and humanized for the receptor (Ces1c/-hDPP4 mice), we...

Response: We thank the review for this comment but since we are limited to 150 words, including this level of detail in the abstract was not possible. Instead, in the introduction we included the verbiage recommended by the reviewer to make clearly describe the mouse model used in our in vivo studies.

3. Introduction, line 12: replace locally with in the Middle East

Response: Thank you. We have removed "locally" and revised that sentence for accuracy.

4. Introduction, last sentence: revise to better describe the animal model. The deletion of the esterase is not new (Sheahan et al., 2017); the new part is that the mouse model has the humanized receptor (Cockrell et al., 2016) and is deleted for the esterase, which allows for the first time the evaluation of these antivirals against MERS-CoV infections in vivo.

Response: Thank you Reviewer #1. We have since added a more detailed description of the model in the introduction. The passage now reads: "Thus, we generated a transgenic mouse with a humanized MERS-CoV receptor (dipeptidyl peptidase 4, hDPP4) and deleted for carboxylesterase 1c (*Ces1c*) to improve the pharmacokinetics of nucleotide prodrugs such that it better approximates the drug exposure profile in humans".

5. Discussion should include some comment about future studies evaluating the emergence of coronavirus drug resistant mutants. Is it likely the same or different for SARS-CoV versus MERS-CoV?

Response: We thank the reviewer for this comment. We have added the following passage into the discussion: "Lastly, we have reported RDV resistance mutations in both mouse hepatitis virus (MHV) and SARS-CoV that shift EC50 values only 3-5-fold⁵¹. Ongoing and future studies are aimed at determining if MERS-CoV is capable of generating resistance to RDV in vitro and whether the mutational spectra are similar to those obtained by MHV and SARS-CoV."

Reviewer #2 (Remarks to the Author):

Major:

1. Viruses designed to express reporter genes are usually highly attenuated. As such, it is not clear how effective RDV would be against the wt virus (as compared to MERS-nLuc). Similarly, the authors use a mouse adapted MERS-CoV in their study, without providing data on the amount and location of adaptive mutations in their virus. It remains unclear whether RDV would be similarly active towards a (human) MERS-CoV isolate.

Response: We thank the reviewer for this requiring some clarity on whether RDV would retain antiviral activity against a wild-type MERS-CoV isolate. Thus, we performed antiviral assays in our resubmission with WT MERS EMC 2012 and our recombinant version of EMC 2012 expressing nLUC. The titers achieved at 24hr post infection in untreated cultures are similar for both viruses (WT mean titer = $1.0E+06$ PFU/mL, nLUC = $2.6E+06$ PFU/mL. The kinetics of the antiviral effect and final EC_{50} values are also similar for both viruses (WT = $0.12 \mu\text{M}$, nLUC = $0.09 \mu\text{M}$). Moreover, our new data is supported by the observation that WT MERS-CoV Jordan strain ($EC_{50} = 0.34$) is also susceptible to the antiviral activity of RDV in an antigen staining/microscopy-based assay (Warren et al, Nature 2016).

Regarding the mouse adapted virus utilized for our studies, this virus was characterized in the 2018 Douglas et. al Virology paper. In addition to citing the Douglas et. al paper, we have included a brief description of the mutations acquired during mouse adaptation in the Materials and Methods.

2. Histologies. Fig 2 E/G: It is impossible to define an Alveolar type I or type II cell on these sections "based on morphology". If the authors want to make this claim, cell-specific markers for these cells need to be applied together with the MERS-Cov Ab.

Response: We thank the reviewer for this comment. MERS-CoV antigen staining of lung sections of human autopsy tissue have demonstrated that both alveolar type I and type II cells are infected by MERS-CoV (Widagdo et. al 2019). Because we have not performed co-staining to definitively prove this targeting in our mouse model, we have altered our language in the results to be less specific and only comment that the antigen positive cells are likely alveolar pneumocytes.

3. It is also not possible to differentiate edema and hyaline membranes at this resolution. In Fig 3 (results text), what is meant by "markers of ARDS" beyond hyaline membranes? Generally, in all the histologies in Fig 2-4, it is surprising that there are few alveolar infiltrates in MERS-CoV infected mice, a hallmark of respiratory virus-induced ARDS. This is particularly low in Fig 3F. I am not sure whether the inflammatory pathology the authors observe in their MERS-CoV model reflects ARDS at all, as the standard criteria are not obvious and it is hard to see from the small sections at the given resolution. Have the sections be analysed by a lung pathologist? Also, the body measurements rather point at an obstructive airway disease, and lack the ARDS-typical reduction of vital capacity and compliance.

Response: We thank the reviewer for highlighting this area needing clarity.

First, we have added a veterinary pathologist (Dr. Stephanie Montgomery, D.V.M., Ph.D.) as an author. Second, Dr. Montgomery has reviewed all of our pathology, and added new photographs describing the pathological features of our treatment groups. There is a consensus among American Thoracic Society in the difficulty in translating the human metrics of acute lung injury (ALI), diffuse alveolar damage (DAD) and acute respiratory distress syndrome (ARDS) to animal models as no animal models recapitulate the pathologic features seen in humans. In 2011 (Maute-Bello et. al) and 2015 (Aefner et. al), a consensus approach to measure ARDS in small animal models was described. Based on those criteria, we find lung histopathologic changes in vehicle treated animals have multiple features of the severe end stage lung disease, acute respiratory distress syndrome (ARDS), including proteinaceous material in alveolar spaces (i.e. evidence of tissue injury), an inflammatory response, and evidence of physiological pulmonary dysfunction. We have created new figures and descriptive

language which describe the various lung pathologic features in greater detail than the original submission.

4. Did therapeutically applied RDV affect survival, as a major translational finding that would allow to move into a clinical trial?

Response: Reviewer #2 raises an important point. We have addressed this request with an additional in vivo study as well as a more adequate explanation of the mouse model as it relates to the human disease. First, for our therapeutic studies reported in the initial submission of the paper, we used a dose of mouse adapted MERS-CoV ($5E+04$ PFU) that rarely causes mortality. Thus, mortality was not an endpoint for these studies. Second, we have included a new therapeutic head to head study comparing therapeutic RDV and LPV/RTV+IFN β initiation one day after MERS-CoV infection with $5E+05$ PFU, which causes mortality. Survival was not different among our groups but only RDV significantly reduced viral loads in the lung. As compared to the human disease which progresses to death or resolves in 3-4 weeks, the kinetics of our mouse model is very compressed. Viral pathogenesis is increased with increasing virus dose. A considerable amount of virus replication occurs within the first 24hr of infection. Thus, even with a potent antiviral such as RDV, viral loads can be reduced by 2 logs with therapeutic treatment yet not improve outcomes due to the replication and associated damage within the first 24hr of infection. We obtained similar data for RDV treatment of SARS-CoV in mice (Sheahan et. al 2017). To date, no animal model faithfully recapitulates all aspects of human MERS-CoV disease. Given that our mouse model captures multiple aspects of the human condition such as high titer replication in the lungs, loss of pulmonary function, lung pathology similar to that seen in severe human MERS-CoV cases, etc., the results we have generated thus far are one of the many complementary pieces of data that would accelerate an antiviral towards human clinical trial.

5. How does a pretreatment with LPV/r (plus and minus IFN) affect titers/outcomes in the mouse model? Only therapeutic treatment data are provided. In this line, the discussion on IFN-beta as a main “driver” of improved outcome in the LPV/r + IFN is in fact not supported by data. LPV/r versus LPV/r + IFN has not been tested in vivo to make this claim. Moreover, there is a lot of evidence in the literature that IFN type I when applied in the course of respiratory viral infection rather drives immunopathology (including lack of barrier function/hemorrhage) and contributes to morbidity in vivo (e.g., PMID 27520969, PMID 28514692), which could explain the data obtained in the LPV/r + IFN groups.

Response: We thank the reviewer for identifying this issue with our data. We have addressed this comment with several new in vivo efficacy studies as well as additional discussion in the text.

First, we chose the LPV/RTV+IFN β regimen and routes of administration to best model the MIRACLE trial ongoing in KSA which is testing the combination of oral LPV/RTV daily and subcutaneous IFN β every other day. While the papers noted above by the Reviewer (Davidson et. al EMBO Mol. Med. 2016, Galani et. al Immunity 2017) show differences Type III vs. Type I interferon in mouse models of influenza, interferons were administered intranasally in both studies, a route which is unusual in humans. Galani et. al also used type I or type III IFN knock out mice to dissect the differences in the host response after influenza infection and determine if the pathways were protective and/or pathogenetic. Galani et. al do show a modest improvement in the outcome of influenza infection following intranasal administration of lambda interferon over vehicle (PBS) although vehicle alone caused significant mortality. Thus, the phenotypes of interferon knock out mice and those administered therapeutic interferon are likely driven by the very different abilities of their innate immune activation. Because of this interesting issue and body of literature we omitted in our discussion, we have added to the discussion to address this important issue.

Second, we performed two additional prophylactic studies aimed at ascertaining whether LPV/RTV+IFN β or IFN β alone exerts an antiviral effect and improves outcomes following MERS-CoV

infection as compared to its vehicle control. In both studies, we also included remdesivir and its appropriate vehicle control. In the first study (Figure S7), for our IFN containing groups, we utilized a 1X human equivalent dose of interferon beta which was initiated 24hr prior to infection and then administered every other day thereafter. In this study, only RDV significantly reduced weight loss and virus lung titers and improved pulmonary function on days 3-6dpi. LPV/RTV+IFNb or IFNb alone did not affect virus lung titers. Mice administered IFNb only lost more weight than both vehicle and LPV/RTV+IFNb suggesting that IFNb alone exacerbates disease. Of the oral groups, only LPV/RTV+IFNb showed improved pulmonary function over vehicle but only on 6dpi. In our pharmacokinetic/pharmacodynamic studies (Figure S8), interferon stimulated gene expression peaked 2-4hr after peripheral administration. Thus, for our second prophylactic study (Figure 3), we mirrored the groups in our therapeutic efficacy studies (Oral Vehicle, LPV/RTV+IFNb low, LPV/RTV+IFNb high) in addition to IFNb high alone but the key difference in this study is that we initiated IFNb 2hr prior to infection and then gave it every other day thereafter. Similar to the first prophylactic study, IFNb alone group exhibited the most weight loss and the worst pulmonary function. While not different from vehicle, LPV/RTV+IFNb low group had significantly improved weight loss as compared to IFNb only. The LPV/RTV+IFNb low group had significantly reduced virus titers as compared to vehicle although the titer reduction was modest (~1/2 log) as compared to that of remdesivir (~3 logs).

Overall, these data suggest that interferon alone provides no improvement of MERS-CoV disease in mice when given prior to infection. Moreover, LPV/RTV+IFNb provides only a modest improvement in outcomes and virus titer reduction as compared to RDV. The addition of these studies as suggested by Reviewer #2 significantly improve our manuscript and help interpret the therapeutic study results.

6. On a minor note, the ARDS trial mentioned here by the authors (ref 32) using type I IFN in ARDS did not prove efficacy).

Response: As we have re-tooled

7. The authors should provide information on the detailed antiviral mechanism of action of RDV in the introduction.

Response: We thank the reviewer for this suggestion and have now included a sentence in the introduction describing the MOA of RDV that was first reported in Warren et. al (Nature 2016).

Reviewers' Comments:

Reviewer #2:

Remarks to the Author:

The manuscript is improved. Major points 1 and 5 have been addressed by providing new datasets. I still have major concerns with the interpretation of the data of both the comparison to the efficacy of therapeutic LPV/r/IFN, and the histology data interpretation with the authors' claim that they ameliorate the "ARDS phenotype" by their treatment.

1) Regarding the regimen comparison between therapeutic RDV (Fig 4) and therapeutic LPV/r/IFN high/low (Fig 5) in the low MERS-CoV dose, it is evident that virus titer reduction was more pronounced in RDV (although, it is nowhere stated at which day data in Fig 5C was taken but I assume d6). As to histologies and "discoloration" score see below. However, lung function was significantly improved in the LPV/r/IFN low group, even at more timepoints than with RDV. So the statement of the authors that "Therapeutic combination of PPV/r/and IFN beta does not.....diminish multiple disease signs" in the Fig 5 legend, or their overall conclusion is not fully supported by data. In the high MERS-Cov infection dose, LPV-containing treatment regimen are even superior in preventing weight loss (Fig S10E), and regarding the survival readout, these data are hard to interpret as in the RDV experiment infection results in a 50% mortality whereas in the LPV-containing regimen it results in a 75% mortality. Again here, the S10 Figure legend states that "Therapeutic RDV ,...but LPV/r/IFN does not affect any metrics following high dose MERS-CoV challenge" which is simply not true. Overall, it is evident that RDV has a much better antiviral effect *in vivo*, but it seems that the authors heavily overstate the superiority of RDV over the LPV regimen regarding lung histologies (below), function, and overall morbidity, that is not supported by data.

2) I have major concerns regarding the histology datasets. First, I definitely do not see an improvement in the RDV group in histologies in Fig 2F (prophylactic) and Fig S4 (here vehicle sometimes looks even better than RDV), as well as Fig 4E and in Fig S9 (unfortunately in these Figs they compare different magnifications, between vehicle and RDV). CD45+ leukocytes are even increased in the high viral dose infection after RDV treatment (which might rather drive than improve morbidity/mortality). The histologies in Fig 5E are again comparing sections of different magnifications and the related Fig S11B (lower magnifications) suggests that the lungs are ameliorated with LPV/RTV plus low IFN β versus vehicle (eventually better than RDV in the other sections). The "perivascular edema" in Fig. S4 (high MERS-CoV dose, vehicle, low magnification) looks like a cutting artefact. The "discoloration" score is very vague (was it assessed in blinded fashion? How was this score performed?) and not a well-accepted lung pathology readout (why do the authors not just show the "discoloured" vs. less discoloured lungs in the figures?).

3) I would also like to point out that the lung function parameters (as outlined in the first comments) do quantify obstructive lung disease, not ARDS (where you would expect reduction in compliance/VC, not in airway resistance). There is improvement of obstruction after the treatment (which is of course supportive), but this is not reflecting an improvement of the phenotype of ARDS that the authors claim to ameliorate with their therapy, and as such its relevance remains a little unclear. A valid "physiological" parameter indicating severe acute lung injury or improvement thereof would be oxygen partial pressure or oxygen saturation as a direct measure of gas exchange (and a parameter defining ARDS in patients).

Reviewer #2 (Remarks to the **Author**):

The manuscript is improved. Major points 1 and 5 have been addressed by providing new datasets.

I still have major concerns with the interpretation of the data of both the comparison to the efficacy of therapeutic LPV/r/IFN, and the histology data interpretation with the authors' claim that they ameliorate the "ARDS phenotype" by their treatment.

1A) Regarding the regimen comparison between therapeutic RDV (Fig 4) and therapeutic LPV/r/IFN high/low (Fig 5) in the low MERS-CoV dose, it is evident that virus titer reduction was more pronounced in RDV (although, it is nowhere stated at which day data in Fig 5C was taken but I assume d6). As to histologies and "discoloration" score see below. However, lung function was significantly improved in the LPV/r/IFN low group, even at more timepoints than with RDV. So the statement of the authors that "Therapeutic combination of PPV/r/and IFN beta does not.....diminish multiple disease signs" in the Fig 5 legend, or their overall conclusion is not fully supported by data.

Response: We unfortunately omitted the timepoint in which virus lung titer was measured in the figure legend and have since corrected this issue. Nevertheless, we provide a new and improved data package to demonstrate that therapeutic RDV improves disease outcomes as compared to LPV/RTV-IFNb.

In the new Figure 5, we show that only therapeutic RDV significantly reduces weight loss, lung hemorrhage, virus titer, and viral antigen in lung tissue sections while LPV/RTV-IFNb does not affect these metrics. In the new Figure 6 comparing pulmonary function for all therapeutic treatments (formerly split among Fig. 4 and 5), we show that both RDV and LPV/RTV-IFN low dose groups can improve lung function. In new Figure 7, we blindly assess and quantitate the histological features of acute lung injury (ALI) using three different and complementary tools which show that only therapeutic RDV reduces the features of ALI. While LPV/RTV-IFNb low improves pulmonary function, only RDV improves multiple disease metrics including those blindly assessed histologic metrics of ALI. Therefore, our claim that therapeutic RDV is superior to LPV/RTV-IFNb is substantiated.

1B) In the high MERS-Cov infection dose, LPV-containing treatment regimen are even superior in preventing weight loss (Fig S10E), and regarding the survival readout, these data are hard to interpret as in the RDV experiment infection results in a 50% mortality whereas in the LPV-containing regimen it results in a 75% mortality. Again here, the S10 Figure legend states that "Therapeutic RDV ,...but LPV/r/IFN does not affect any metrics following high dose MERS-CoV challenge" which is simply not true.

Response: This reviewer brings up a good point regarding the language in the legend title of Figure S10 (New Fig. S6). Since we found no improvement in survival among our treatment groups, we have changed the presentation and discussion of this figure and

now write that “no treatment improved survival or lung hemorrhage but therapeutic RDV significantly reduced lung viral load on 6dpi”. As we found with our SARS-CoV studies (Sheahan et al. 2017), the degree of clinical benefit is dependent on the viral burden and time of treatment initiation. Thus, the damage inflicted during the initial 24hr of infection with the lethal dose of MERS-CoV was too extensive for the therapeutic treatments to overcome.

1C) Overall, it is evident that RDV has a much better antiviral effect in vivo, but it seems that the authors heavily overstate the superiority of RDV over the LPV regimen regarding lung histologies (below), function, and overall morbidity, that is not supported by data.

Response: We agree with the reviewer that RDV has a much better antiviral effect in vivo. A direct reduction in virus replication is the initial metric that all antiviral therapies are measured against since diminishing viral load is usually the first step in ameliorating viral disease. We found that only prophylactic LPV/RTV IFN β minimally affected viral and therapeutic treatment did not affect virus titer. Therefore, only therapeutic RDV satisfied this very basic metric of antiviral efficacy. To improve the histological data in our manuscript, we blindly assessed and quantitated the histological features of acute lung injury (ALI) using three different and complementary tools which show that only therapeutic RDV reduces the features of ALI. While LPV/RTV-IFN β low improves pulmonary function, only RDV improves multiple disease metrics including those blindly assessed histologic metrics of ALI. Therefore, our claim that therapeutic RDV is superior to LPV/RTV-IFN β is substantiated.

2) I have major concerns regarding the histology datasets.

2A) First, I definitely do not see an improvement in the RDV group in histologies in Fig 2F (prophylactic) and Fig S4 (here vehicle sometimes looks even better than RDV), as well as Fig 4E and in Fig S9 (unfortunately in these Figs they compare different magnifications, between vehicle and RDV).

Response: The improvement in the histological features of acute lung injury (ALI) with RDV treatment is difficult to appreciate at low magnification. In collaboration with a board certified veterinary pathologist, we blindly reassessed and quantitated the histological features of ALI in the new Figure 3 (formerly Fig. 2 and S4) and new Figure 7 (formerly Fig. 4, 5, S9 and S11). It is recognized by groups like the American Thoracic Society (ATS) that direct translation of the measurements and metrics of human ALI are challenging and often not appropriate in animal models of ALI. As a result, ATS issued a consensus document with defined parameters and tools to simplify the translation of mouse models of ALI to the human condition. Thus, using a histological assessment tool for ALI developed by the ATS as well as two other complementary and quantitative approaches, we found that only therapeutic RDV diminished the histologic features of ALI (new Fig. 7). Where the magnification and presentation of histology in former Figure 2F made it difficult for Reviewer #2 to appreciate differences among vehicle and treated groups, we also now show clear, quantitative and significant differences among vehicle

groups and those administered RDV prophylaxis in new Figure 3. All together, we show through three different, complementary and blinded approaches that only therapeutic RDV significantly diminished histological features consistent with ALI. Therefore, our claim that therapeutic RDV is superior to LPV/RTV-IFN β is substantiated.

2B) CD45+ leukocytes are even increased in the high viral dose infection after RDV treatment (which might rather drive than improve morbidity/mortality).

Response: We have removed these and related data in order to keep the current manuscript focused and more concise.

2C) The histologies in Fig 5E are again comparing sections of different magnifications and the related Fig S11B (lower magnifications) suggests that the lungs are ameliorated with LPV/RTV plus low IFN β versus vehicle (eventually better than RDV in the other sections).

Response: We have retaken all the photographs of histology such that all magnifications are consistent (60X objective). We now show through three different, complementary and blinded approaches that only therapeutic RDV significantly diminished histological features consistent with ALI. Therefore, our claim that therapeutic RDV is superior to LPV/RTV-IFN β is substantiated. See the above rebuttal to point 2A for more information.

2D) The “perivascular edema” in Fig. S4 (high MERSCoV dose, vehicle, low magnification) looks like a cutting artefact.

Response: Since perivascular edema was not part of our quantitative scoring schemes for acute lung injury, we have removed mention of this phenotype from the current manuscript. Nevertheless, a “perivascular edema” -like phenotype can be caused by a cutting artifact or incorrect perfusion in lung tissue, but the presence of inflammatory cells in the edema space strongly suggested that our phenotype was not artifactual.

2F) The “discoloration” score is very vague (was it assessed in blinded fashion? How was this score performed?) and not a well-accepted lung pathology readout (why do the authors not just show the “discoloured” vs. less discoloured lungs in the figures?).

Response: We apologize for this issue of confusing nomenclature. Gross macroscopic changes are associated with severe lung damage and this among other metrics was noted as a feature of acute lung injury in mice by the American Thoracic Society in 2010 (Matute-Bello et. al 2010, Aefner and Davis 2015). While we do not have photos of this gross pathological phenotype, we have been measuring this phenotype in both SARS- or MERS-CoV mouse models for over a decade as “hemorrhage” score. In our manuscript, the scientists dissecting animals were not blinded to treatment at the time of sacrifice therefore this metric was not assessed blindly. We have changed the nomenclature in the current manuscript from “discoloration” to “hemorrhage” to be

consistent with our previous publications (2013 Gralinski et al., 2016 Cockrell et al.). Of note, this same phenotype has been reported by unrelated groups in mouse models of influenza pathogenesis (2011 Fukushi et al.).

3) I would also like to point out that the lung function parameters (as outlined in the first comments) do quantify obstructive lung disease, not ARDS (where you would expect reduction in compliance/VC, not in airway resistance). There is improvement of obstruction after the treatment (which is of course supportive), but this is not reflecting an improvement of the phenotype of ARDS that the authors claim to ameliorate with their therapy, and as such its relevance remains a little unclear. A valid “physiological” parameter indicating severe acute lung injury or improvement thereof would be oxygen partial pressure or oxygen saturation as a direct measure of gas exchange (and a parameter defining ARDS in patients).

Response: We have removed all mentions of ARDS in our current manuscript in favor of the similar yet more easily defined phenotype of acute lung injury (ALI). The phenotypes, physiology and metrics describing ALI/ARDS in humans as defined by the American European Consensus Conference cannot be directly extended to models of severe lung disease in mice. As suggested by Review #2, we cannot obtain the physiological measure of partial pressure of oxygen (PaO₂) in our MERS-CoV model. This measurement would require special equipment and training (i.e. blood gas analyzer, ventilation, intubation, etc.) (Traylor et. al 2012) common in a hospital setting, but not available or allowed in our Biosafety Level 3 (BSL3) due to the risk of aerosol generation. In the past, we have attempted to correlate the less invasive pulse oximetry (i.e. blood oxygen saturation) with SARS-CoV pathogenesis but could not obtain consistent data using the tools available at BSL3. For these reasons, we use whole body plethysmography (WBP) to assess pulmonary function which we and others have shown correlates well with the histopathologic features of ALI in mouse models of severe coronavirus (Cockrell et al. 2016) and respiratory syncytial virus infection (Schmidt et. al 2018). Thus, WBP can be used to assess pulmonary function following virus infection and changes in WBP metrics are associated with histopathological features consistent with acute lung injury.

Aside from the practical issues of studying ALI in humans and mice, differences in physiology, anatomy, biology and immunology make the direct translation of human disease criteria to mouse models complicated. Because of this, the American Thoracic Society produced a consensus document describing the main features and measurements of ALI in mouse models providing a framework for the translation of these models to the human condition. Thus, according to the ATS, there are several “valid” features of ALI that can be measured in small animal models using various experimental approaches and tools. Using the ATS acute lung injury scoring tool, we blindly scored the histopathologic features of ALI in lung tissue sections. We provide conclusive quantitative data in our latest manuscript draft showing that only therapeutic Remdesivir reduced the histologic features of ALI. In addition, we found similar results using a complementary histologic scoring tool for diffuse alveolar damage (2018 Schmidt et al.). Lastly, in the same tissues scored for lung injury, we quantitated cleaved caspase-3 antigen staining and determined that only therapeutic Remdesivir

significantly reduced cell death. Thus, with three different blinded quantitative approaches, we show that therapeutic Remdesivir provides superior protection from lung injury to LPV/RTV-IFN β . Therefore, our claim that therapeutic RDV is superior to LPV/RTV-IFN β is substantiated.

Reviewers' Comments:

Reviewer #2:

Remarks to the Author:

The manuscript is substantially improved and the new histology quantifications are very helpful in better estimating the effects of RDV versus LPV/r in this regard.

Regarding the mortality data (see previous comment 1b in the rebuttal letter) it is still quite surprising that, although there's a substantial decrease in viral load, no survival benefit occurs. Given that the authors reveal a substantial reduction in viral load of 4 logs at 6 dpi in Fig 5D, this should also reduce mortality (as seemingly it also affects body weight). It likely depends on an appropriate dosing/timing of virus inoculum and RDV, but I think this should be shown to support the relevance for human infection and putative clinical implications, and – the major claim of the authors – superiority over LPV/r/IFN treatment. The authors mentioned that they “found with their SARS-CoV studies (Sheahan et al. 2017), that the degree of clinical benefit is dependent on the viral burden and time of treatment initiation. Thus, the damage inflicted during the initial 24hr of infection with the lethal dose of MERS-CoV was too extensive for the therapeutic treatments to overcome”. If only very early treatment is successful, the clinical relevance of the findings on therapeutic RDV treatment is in fact questionable, given that therapeutic treatment would only be initiated after start of significant symptoms, which is likely not during the very first day of infection.

Just on a minor note, PaO₂ measurement does not require ventilation and intubation, and can be performed in mice breathing room air (FiO₂ 21%) from arterial blood eg by left heart puncture (these experiments are no longer required as the authors do no longer state that they ameliorate ARDS, and the new histology analyses are very supportive with regard to RDV effects on ALI).

Reviewer #2 (Remarks to the Author):

The manuscript is substantially improved and the new histology quantifications are very helpful in better estimating the effects of RDV versus LPV/r in this regard.

Regarding the mortality data (see previous comment 1b in the rebuttal letter) it is still quite surprising that, although there's a substantial decrease in viral load, no survival benefit occurs. Given that the authors reveal a substantial reduction in viral load of 4 logs at 6 dpi in Fig 5D, this should also reduce mortality (as seemingly it also affects body weight). It likely depends on an appropriate dosing/timing of virus inoculum and RDV, but I think this should be shown to support the relevance for human infection and putative clinical implications, and – the major claim of the authors – superiority over LPV/r/IFN treatment. The authors mentioned that they “found with their SARS-CoV studies (Sheahan et al. 2017), that the degree of clinical benefit is dependent on the viral burden and time of treatment initiation. Thus, the damage inflicted during the initial 24hr of infection with the lethal dose of MERS-CoV was too extensive for the therapeutic treatments to overcome”. If only very early treatment is successful, the clinical relevance of the findings on therapeutic RDV treatment is in fact questionable, given that therapeutic treatment would only be initiated after start of significant symptoms, which is likely not during the very first day of infection.

Just on a minor note, PaO₂ measurement does not require ventilation and intubation, and can be performed in mice breathing room air (FiO₂ 21%) from arterial blood eg by left heart puncture (these experiments are no longer required as the authors do no longer state that they ameliorate ARDS, and the new histology analyses are very supportive with regard to RDV effects on ALI).

Response: This comment is directed towards our data showing that we see reductions in viral load with lethal challenge in animals treated with remdesivir (Supplementary Figure 6) but do not prevent mortality. Similarly, infected animals treated with LPV/RTV+IFN β do not show reductions in viral load or mortality. As mouse models of MERS-CoV pathogenesis do not mirror all aspects of the human disease, we dedicate an entire paragraph of the Discussion to explain this issue including how the disease course in mice is significantly compressed as compared to that in humans. In addition, to cause consistently lethal disease in the mouse model, a significant amount of virus must be administered ($5E+05$ pfu) in a single bolus which is able to simultaneously act and infect the airway epithelium in concert infecting many many cells at once. This is likely not how human infection occurs or progresses. Because of the differences of MERS-CoV disease humans and mice, failure to protect mice from lethal challenge with therapeutic treatments is not necessarily indicative of imminent clinical trial failure in humans and is more a reflection on the idiosyncrasies of mouse model. We thank the review for the comment on the new histological analysis!